# Enhancing the Cross-Size Generalization for Solving Vehicle Routing Problems via Continual Learning

## Abstract

Deep models for vehicle routing problems are typically trained and evaluated using instances of a single size, which severely limits their ability to generalize across different problem sizes and thus hampers their practical applicability. To address the issue, we propose a continual learning based framework that sequentially trains a deep model with instances of ascending problem sizes. Specifically, on the one hand, we design an inter-task regularization scheme to retain the knowledge acquired from smaller problem sizes in the model training on a larger size. On the other hand, we introduce an intra-task regularization scheme to consolidate the model by imitating the latest desirable behaviors during training on each size. Additionally, we exploit the experience replay to revisit instances of formerly trained sizes for mitigating the catastrophic forgetting. Extensive experimental results show that the proposed approach achieves predominantly superior performance across various problem sizes (either seen or unseen in the training), as compared to state-of-the-art deep models including the ones specialized for the generalizability enhancement. Meanwhile, the ablation studies on the key designs manifest their synergistic effect in the proposed framework.

## 1 Introduction

Vehicle routing problems (VRPs), including the Traveling Salesman Problem (TSP) and the Capacitated Vehicle Routing Problem (CVRP), aim to find the optimal route for vehicles serving a group of customers in various real-life scenarios, such as parcel pickup/delivery, passenger transportation, and home health care (Baker & Ayechew, 2003; Schneider et al., 2014). Despite the extensive efforts in computer science and operations research, traditional exact and heuristic algorithms still encounter challenges when solving VRPs due to their NP-hard nature (Lenstra & Kan, 1981). These algorithms often require massive tuning to determine the hand-crafted rules and related hyperparameters. To mitigate this issue, deep (reinforcement) learning based methods have been extensively studied and applied to solve VRPs in recent years (Bengio et al., 2021; Mazyavkina et al., 2021; Zhang et al., 2023a), which leverage neural networks to automatically learn (heuristic) policies from the experience of solving similar VRP instances. Bolstered by advanced neural networks and training approaches, some of these deep models have achieved competitive or even superior performance to the traditional algorithms (Kwon et al., 2020; Ma et al., 2021; Li et al., 2023).

Typically, existing deep models are often trained and evaluated on single-sized problem instances, where they are able to deliver decent and efficient solutions. However, the performance of learned policies diminishes when applied to sizes not encountered during the training phase. This limitation becomes more pronounced as the disparity between the sizes further increases. Such a cross-size generalization issue considerably hinders the applications of deep models, especially given that real-world VRP instances consistently present a diverse range of problem sizes.

To address this issue, we propose a continual learning (CL) (Chen & Liu, 2018) based framework that sequentially trains a deep model on instances of ascending problem sizes. This approach enables the model to perform favorably across a range of problem sizes, covering both those seen and unseen during the training phase. Specifically, we preserve exemplary models derived from the previous training, and leverage the regularization scheme to retain their knowledge for facilitating

the subsequent model training. We design two distinct regularization terms in the loss function, i.e., the inter-task and the intra-task regularization terms. During the training on each size, the former aims to transfer the valuable insights from smaller-sized tasks to larger-sized ones, while the latter enables the imitation of the most recent exemplar models. Intuitively, both schemes expedite the training on a newly encountered size, with the aid of previously attained experience in problem solving. Additionally, we tailor an experience replay technique (Rolnick et al., 2019) to intermittently revisit the training instances of smaller sizes (trained on previously), with the intent of mitigating the catastrophic forgetting (French, 1999) inherent in continual learning. Notably, the proposed continual learning only improves the training algorithm of existing deep models, without altering their original neural architectures. It has a great potential to be deployed with different models, without inducing extra inference time. Experimental results indicate that our approach significantly raises the cross-size generalization performance of deep models for both seen and unseen problem sizes. Furthermore, it generally outperforms the state-of-the-art methods that are specialized for enhancing the generalizability of deep models, showing the effectiveness of our algorithmic designs.

Accordingly, our contributions are summarized as follows: (1) We propose a model-agnostic continual learning based framework to improve the cross-size generalization capabilities of deep models for VRPs. With a single training session, the proposed approach empowers deep models to deliver promising results for vehicle routing across a wide range of problem sizes, without incurring extra inference time. (2) To expedite the training on new sizes, we design the inter-task regularization scheme to facilitate the knowledge transfer from smaller to larger problem sizes. Alternatively, the intra-task regularization scheme consolidates the model by imitating the most recent exemplar models on the current size. On the other hand, we employ the experience replay to counteract the catastrophic forgetting, retaining the competence of deep model in handling smaller-size instances beyond its training on larger ones. (3) We evaluate our approach on TSP and CVRP, across a broad spectrum of sizes (seen or unseen during the training). Results on both synthetic and benchmark datasets show that our approach bolsters the cross-size generalization, yielding predominantly superior performance to the state-of-the-art methods specialized for generalizability enhancement.

## 2 RELATED WORK

In this section, we review deep models for VRPs and representative works on enhancing cross-size generalization. Then, we brief on the generic continual learning in the machine learning community.

**Deep models for VRPs.** Recent learning based methods, i.e., deep models, have shown promise in solving VRPs by automatically discovering effective policies. Vinyals et al. (2015) tendered the Pointer network to learn constructing TSP solution supervisedly, which was further extended to reinforcement learning (Bello et al., 2017) and CVRP (Nazari et al., 2018). Similarly, the graph conventional network (GCN) was leveraged to estimate probabilities of each edge appearing in the optimal TSP solution (Joshi et al., 2019). With recent advances of the self-attention mechanism, the attention model (AM) (Kool et al., 2018) was tailored from Transformer (Vaswani et al., 2017) for solving VRPs and recognized as a landmark contribution in this field. The follow-up works diverged by (slightly) restructuring AM or targeting diverse VRP variants (Xin et al., 2020; Li et al., 2021a). The policy optimization with multiple optima (POMO) (Kwon et al., 2020) improved AM by exploiting symmetric rollouts and data augmentation technique, achieving state-of-the-art performance for VRPs. Despite the efficient inference, the above methods usually require heavy post-processing procedures to enhance solution quality, such as sampling (Li et al., 2021b), active search (Hottung et al., 2021). Especially, some works attempt to improve the generalization performance of deep models in handling distribution shift (Jiang et al., 2022; Wang et al., 2022; Bi et al., 2022; Hottung et al., 2021; Zhou et al., 2023). Instead, this paper aims to enhance the cross-size generalization towards a deep model capable of well solving different-sized VRPs.

**Cross-size generalization.** The above deep models are often trained to solve single-sized VRP instances for attaining favorable evaluation results on that problem size. However, their performance degenerates when the models are evaluated on sizes unseen during the training. To address this cross-size generalization issue, Lisicki et al. (2020) proposed a curriculum learning method to solve TSP instances spanning a range of problem sizes. Similarly, Zhang et al. (2023b) utilized the curriculum learning to train a deep model on different-sized TSP, with the knowledge distillation used for

training on the largest TSP. Nevertheless, both methods are limited to TSP and lack the versatility in addressing broader VRP variants. Instead, Zhou et al. (2023) worked on improving generalization performance across sizes and distributions, by introducing a meta-learning approach to initialize deep models for rapid adaptation to target VRPs. However, its performance is contingent on the heavy base model and tricky meta-learning process, which could suffer from a high training cost in the absence of well pre-trained deep models.

In this paper, we first use continual learning to enhance cross-size. Note that our work is different from the ones attempting to solve large-scaled VRPs, which require extra inefficient training/post-processing for the target size Qiu et al. (2022); Sun & Yang (2023); Li et al. (2021c); Fu et al. (2021); Hou et al. (2022); Zong et al. (2022). Our overarching goal is developing a single model with favorable performance in a broad spectrum of problem sizes, in only a single training session.

**Continual learning.** Continual learning (CL) is advantageous in sequentially learning a stream of relevant tasks by absorbing and accumulating knowledge over them Hadsell et al. (2020). However, CL is generally limited by catastrophic forgetting, where learning a new task usually results in a performance degradation on the old tasks. To address this issue, numerous efforts have been devoted in recent years to strike a desirable balance between learning plasticity and memory stability. These works can be broadly categorized into three groups, i.e., regularization-based approaches (Li & Hoiem, 2017) that regularize the current training with the knowledge acquired in the past training; replay-based approaches (Rebuffi et al., 2017) that revisit data distributions of previous tasks; and parameter isolation approaches (Mallya & Lazebnik, 2018) that freeze parameters associated with earlier tasks. Continual learning has widespread applications in visual classification (He & Zhu, 2021), semantic segmentation (Michieli & Zanuttigh, 2019), natural language processing (Han et al., 2021), to name a few. We direct interested readers to De Lange et al. (2021); Parisi et al. (2019) for more details of CL. In this paper, we introduce the continual learning into VRP domain, and empirically testify its potential in training deep models that favorably solve different-sized VRPs.

## 3 PRELIMINARIES AND NOTATIONS

We first formally describe the vehicle routing problems (VRPs) with the objective of yielding high-quality solutions across a spectrum of problem sizes. Then, we present the commonly used encoder-decoder structured deep models for constructing solutions to VRPs in an autoregressive manner.

### 3.1 VRP STATEMENT

Following the literature (Kool et al., 2018; Wu et al., 2021), we focus on two representative routing problems, i.e., TSP and CVRP, respectively. We define a VRP instance over a graph $G = (V, E)$, where $V$ signifies (customer) nodes and $E$ signifies edges between every two different nodes. With $N$ customer in different locations, TSP aims to find the shortest Hamiltonian cycle of $V = \{v_i\}_1^N$, which satisfies that each node in $V$ is visited exactly once. With an auxiliary depot node $v_0$, CVRP extends TSP by considering a fleet of identical vehicles, each of which traverses locations of customers for serving them. Specifically, each vehicle starts from the depot, serves a subset of customers and ultimately returns to the depot. The constraint on the route of a vehicle is that the total demand of customers in a route cannot exceed the vehicle capacity and each customer is visited exactly once.

**Objective Function.** The solution (i.e., tour) $\tau^N$ to a VRP instance can be described as a permutation of $N$ nodes in $V$. The objective function is often defined as the tour length. For example, the objective function of TSP is $C(\tau^N) = \sum_{\{v_i, v_j\} \in \tau^N} D(v_i, v_j)$, where $D(v_i, v_j)$ means the Euclidean distance between the nodes $v_i$ and $v_j$. In this paper, we focus on optimizing objective values of VRPs across multiple problem sizes. By referring various sizes to a series $\{N_1, N_2, ..., N_K\}$, the cross-size objective function could be defined as the average value of the expected tour lengths over the $K$ sizes, i.e., $L = \frac{1}{K} \sum_{i=1}^{K} \mathbb{E}[C(\tau^{N_i})]$, reflecting the overall performance of deep models.

### 3.2 AUTOREGRESSIVE DEEP MODELS FOR VRPS

Deep models often learn constructing solutions to TSP instances in an autoregressive manner. Specifically, they model the solution construction procedure of VRPs as a Markov Decision Pro-

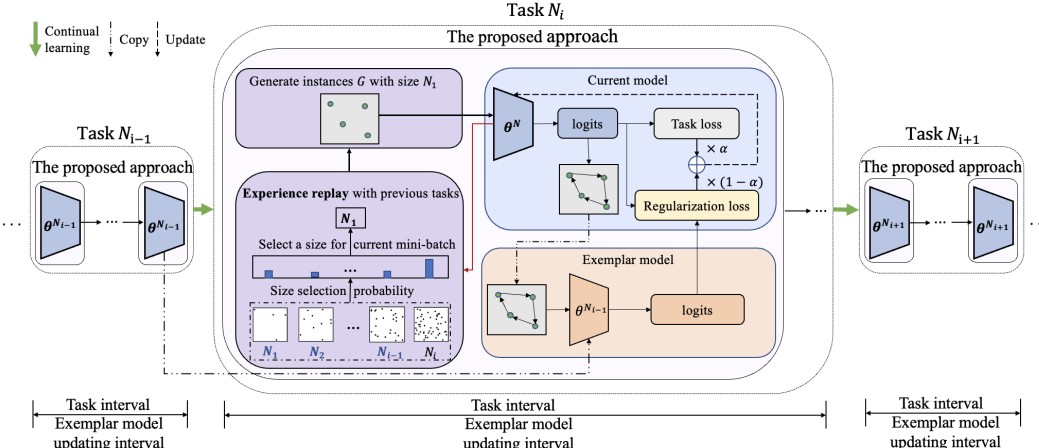

Figure 1: The illustration of the proposed framework with inter-task regularization. For each mini-batch training during current task interval, we employ 1) experience replay to sample a size from formerly trained sizes and current one, and generate instances with that sampled size; 2) inter-task regularization to foster the current model to emulate an exemplary model for knowledge retention.

cess (MDP). Then the encoder-decoder structured policy network is adopted to sequentially construct solutions. More specific, the encoder projects problem-specific features into high-dimensional node embeddings for informative representation learning. Afterwards, the decoder sequentially constructs a solution $\tau^{N_i}$ for a TSP instance of problem size $N_i$, conditioned on the updated node embeddings and partial tour at each step. During solution construction, the decoder selects a node $a_{t_c}$ at step $t_c$, with all constraints satisfied by masking the invalid nodes. A feasible solution is constructed until all customer nodes are selected, which is expressed by the factorization below,

$$p_\theta(\tau^{N_i}|G) = \prod_{t_c=1}^{T_c} p_\theta(a_{t_c}|a_{1:t_c-1}, G), \tag{1}$$

where $p_\theta$ and $T_c$ signifies the policy network and the total number of decoding steps, respectively. In particular, $T_c = N_i$ for TSP, and $T_c \geq N_i$ for CVRP as the depot node can be visited multiple times.

## 4 METHODOLOGY

Continual learning has emerged as a powerful approach for handling sequential tasks, which enables deep models to progressively retain and accumulate knowledge from evolving data streams. As illustrated in Figure 1, we harness CL to enhance the cross-size generalization capability of an autoregressive deep model $\theta$ (e.g., POMO (Kwon et al., 2020) (see Appendix E for details)), by sequentially training it on VRP instances of ascending problem sizes $\{N_1, N_2, ..., N_K\}$. To ensure general favorable performance across the size spectrum, each size (i.e., task) $N_i$ ($i = 1, ..., K$) is considered equally important and trained with the same task interval, which is defined as $E_p = E/K$ epochs where $E$ denotes the total training epochs of CL. In each task interval, the model is trained on each size to optimize the task-specific objective. Meanwhile, our approach exploits experience replay strategy to revisit instances of previously trained smaller sizes, so as to mitigate the catastrophic forgetting. Moreover, the inter-task or intra-task regularization scheme foster the model in current interval to emulate an exemplary model derived from previous or current interval, so as to inherit the previous learned knowledge. In this sense, our CL approach facilitates a coherent continuum of learning across varying problem sizes, which is elaborated in the following sections.

### 4.1 EXPERIENCE REPLAY

Experience replay has shown promise to alleviate the catastrophic forgetting issue in continual learning, with the basic logic of reminding the model about the policy learned for previous tasks. A typical experience replay technique is to maintain a small memory buffer of training samples. These samples are collected from the past tasks and replayed during the training on subsequent tasks. Given

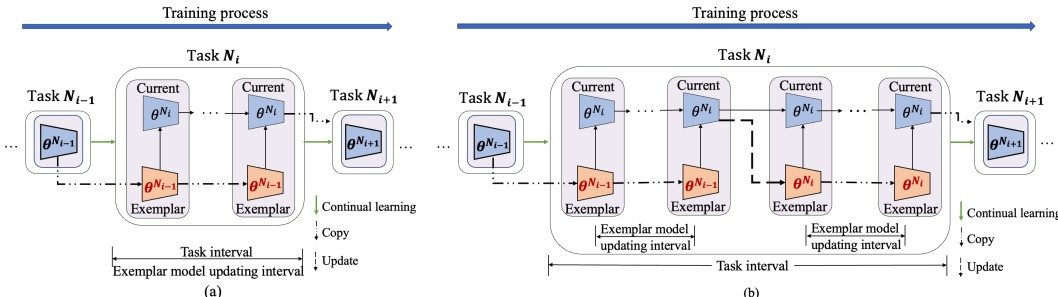

Figure 2: Regularization with two exemplar model updating strategies. (a) inter-task: exemplar model is updated after training on a whole task; (b) intra-task: exemplar model is updated multiple times during training on a task for concentrating more on newly encountered (larger) size.

that existing deep models for VRPs are generally trained with random instances (Kool et al., 2018; Kwon et al., 2020), we propose to randomly generate instances of smaller sizes on the fly. Such real-time memory buffer is able to reflect the instance patterns in previous tasks and raise the memory efficiency, when the deep model is trained on a newly encountered larger size.

During the training on the problem size $N_i$ ($i > 1$), we harness a sampling strategy to either randomly select a size from the set of formerly trained sizes $N_{pre} = \{N_1, ..., N_{i-1}\}$, or deterministically select the current size $N_i$. This strategy is devised to ensure that the deep model is primarily trained on the current task, i.e., the VRP with a larger size and higher complexity than the previous ones. Meanwhile, it ensures the competence of the deep model is retained for well solving previous tasks, i.e., the VRPs with smaller sizes but subjected to the catastrophic forgetting. To this end, we sample problem sizes in mini-batches during the training on size $N_k$, by assigning a higher probability to select $N_i$ and a lower probability to uniformly select one from $N_{pre}$, such that,

$$N_k = \begin{cases} N_i, & \text{if } \epsilon < 0.5 \\ N_j \sim U(N_{pre}), & \text{otherwise} \end{cases} \tag{2}$$

where $\epsilon \in (0, 1)$ is a random number. Specially, only the size $N_1$ is involved in the first task.

## 4.2 REGULARIZATION SCHEMES

During the training process, we employ favorable models trained previously as the *exemplar* ones to infuse the *current* model with a wealth of knowledge in VRP solving, with the goal to guide the training on the newly encountered size. Specifically, we design two distinct terms in the loss function, i.e., inter-task regularization term and intra-task regularization term, respectively, with different update rule for the exemplar model. Note that only one regularization scheme can be used in our CL approach to keep a stable update of the exemplar model throughout the training.

**Inter-task regularization scheme.** As shown in Figure 2(a), the inter-task regularization scheme aims to retain knowledge derived from the past training on smaller sizes for achieving generalization across various sizes. Specifically, when training on size $N_i$, the current model $\theta^{N_i}$ is thoughtfully guided by the exemplar model $\theta^{N_{i-1}}$ meticulously trained on the preceding size $N_{i-1}$. In this fashion, the exemplar model is updated after training on each size, with the update interval equal to the task interval, i.e., $E_{inter} = E_p$. This strategy encourages the current model to imitate the solution construction policy learned by the exemplar model. Given a training instance $G$ with size $N_i$ and a tour $\tau_{\theta^{N_i}}$ constructed by $\theta^{N_i}$, we leverage the exemplar model $\theta^{N_{i-1}}$ to engender the same tour, resulting in the probability distribution $p_{\theta^{N_{i-1}}}(\tau_{\theta^{N_i}}|G)$. The inter-task regularization loss $\mathcal{L}_{R_{inter}}$ is defined as the similarity between probability distributions derived by $\theta^{N_i}$ and $\theta^{N_{i-1}}$ over a mini-batch of instances $\{G_b\}_{b=1}^B$, which is calculated by the Kullback-Leibler divergence as below,

$$\mathcal{L}_{R_{inter}} = \frac{1}{B} \sum_{b=1}^{B} \sum_{a_j \in \tau_{\theta^{N_i}}^b} p_{\theta^{N_{i-1}}}(a_j|G_b)(log p_{\theta^{N_{i-1}}}(a_j|G_b) - log p_{\theta^{N_i}}(a_j|G_b)). \tag{3}$$

Particularly, for training on the first size, a pre-trained backbone model (such as POMO (Kwon et al., 2020)) on size $N_1$ could be used to serve as the exemplar model in Eq. (3).

---

Algorithm 1: Model training by continual learning

---

**Input:** An ascending sequence of problem sizes $N_1, N_2, ..., N_K$ with equal space $n$; a pre-trained backbone model (e.g., POMO) parameterized by $\theta^{N_1}$ on size $N_1$;

1: **for** epoch $e = 1, 2, ..., E$ **do**
2:     Compute the size $N_i = N_1 + n * (e \% E_p)$ of current task;
3:     **for** step $t = 1, 2, ..., T$ **do**
4:         Pick a size $N_k, k = 1, ..., i$ according to Eq. (2);
5:         Randomly generate a batch of training instances with size $N_k$;
6:         Let model $\theta$ (e.g., $\theta^{N_i}$ for inter-task regularization) sample tours $\tau_\theta^b$ for each $\{G_b\}_{b=1}^B$;
7:         Compute $\nabla \mathcal{L}_R$ using Eq. (3) for inter-task regularization or Eq. (4) for intra-task one;
8:         Compute $\nabla \mathcal{L}_T$ using Eq. (**??**);
9:         $\theta \leftarrow \theta + \eta \nabla \mathcal{L}$ where $\nabla \mathcal{L} \leftarrow \alpha \nabla \mathcal{L}_R + (1 - \alpha) \nabla \mathcal{L}_T$.
10:    **end for**
11: **end for**

---

**Intra-task regularization scheme.** As illustrated in Figure 2(b), the intra-task regularization scheme concentrates more on consolidating the recently learned knowledge, thereby updating the exemplar model more frequently than inter-task scheme. Specifically, during the training on size $N_i$ in the task interval, we update the exemplar model $M$ times with an even update interval $E_{intra} = E_p/M$. Given the current epoch $e$, the training of the model $\theta_e^{N_i}$ is guided by the most recent exemplar model $\theta_m^{N_i}$ ($m = 1, 2, ..., M$). Accordingly, the intra-task regularization loss $\mathcal{L}_{R_{intra}}$ over a mini-batch of instances $\{G_b\}_{b=1}^B$ is formulated as follows,

$$\mathcal{L}_{R_{intra}} = \frac{1}{B} \sum_{b=1}^B \sum_{a_j \in \tau_{\theta_e^{N_i}}^b} p_{\theta_m^{N_i}}(a_j|G_b)(log p_{\theta_m^{N_i}}(a_j|G_b) - log p_{\theta_e^{N_i}}(a_j|G_b)). \tag{4}$$

In contrast to inter-task regularization scheme using exemplar model from previous size, intra-task scheme adopts the one that has already been exposed to the intricacies of a new size, which could assimilate more generalized and resilient knowledge to boost the training efficiency and is preferred for generalizing to unseen larger sizes. However, the deep model cannot be sufficiently trained on a new size in the initial stage of a task interval. Thus we employ the finally well-established model $\theta_M^{N_{i-1}}$ on the last size as the exemplar, during the first $E_{intra}$ epochs in the current task interval.

Finally, the deep model is trained with the objective of minimizing a weighted combination of the regularization term $\mathcal{L}_R$ (i.e., $\mathcal{L}_{R_{inter}}$ for inter-task regularization and $\mathcal{L}_{R_{intra}}$ for intra-task regularizatio) and the original task loss $\mathcal{L}_T$ as below,

$$\mathcal{L} = \alpha \mathcal{L}_R + (1 - \alpha) \mathcal{L}_T, \tag{5}$$

where $\alpha \in [0, 1]$. Taking inter-task regularization term as an example, the task loss is formulated as $\mathcal{L}_T = \mathbb{E}_{G \sim N_k, \tau^{N_k} \sim p_{\theta^{N_i}}(\tau^{N_k}|G)}[C(\tau^{N_k}|G)]$, where the training instances are sampled with the selected size $N_k$ via the experience replay strategy, and the tour $\tau^{N_k}$ is engendered via the current network $\theta^{N_i}$ according to Eq. (1). The task loss is used to update the deep model by REIN-FORCE (Williams, 1992), which is a commonly applied reinforcement learning algorithm in VRP literature Kool et al. (2018); Kwon et al. (2020).

## 4.3 TRAINING ALGORITHM

We outline the training procedure of the proposed CL approach in Algorithm 1, where the model is sequentially trained using instances with ascending problem sizes $N_1, ..., N_K$. Particularly, starting with the training on size $N_2$, the experience replay strategy plays the role to retain the competence in tackling smaller-size instances when addressing a new larger one. Moreover, the regularization scheme, i.e., either inter-task or intra-task, is smoothly incorporated during the whole training process, transferring previous valuable knowledge to facilitate the subsequent training. In this sense, the proposed approach is expected to endow the deep models with strong cross-size generalization ability so that they could perform favorably across a wide range of sizes.

Table 1: Comparison results on TSP and CVRP (seen scales).

| | Method | Test on N=60 | | | Test on N=100 | | | Test on N=150 | | | Average of Total costs |
|---|---|---|---|---|---|---|---|---|---|---|---|
| | | Obj. | Gap | Time# | Obj. | Gap | Time# | Obj. | Gap | Time# | |
| TSP | Concorde | 6.1729 | - | (7m) | 7.7646 | - | (1.7h) | 9.3462 | - | (22m) | 7.7612 |
| | LKH3 | 6.1729 | 0.00% | (14m) | 7.7646 | 0.00% | (9.8h) | 9.3462 | 0.00% | (2.1h) | 7.7612 |
| | AMDKD-POMO* | 6.1828 | 0.16% | 36s | 7.7930 | 0.37% | 2m | 9.4539 | 1.15% | 33s | 7.8092 |
| | POMO-60 | **6.1746** | **0.03%** | ~ | 7.8050 | 0.52% | ~ | 9.5909 | 2.62% | ~ | 7.8568 |
| | POMO-100 | 6.1768 | 0.06% | ~ | **7.7753** | **0.14%** | ~ | 9.3987 | 0.56% | ~ | 7.7836 |
| | POMO-150 | 6.1928 | 0.32% | ~ | 7.7875 | 0.30% | ~ | **9.3812** | **0.36%** | ~ | 7.7868 |
| | POMO-random | 6.1778 | 0.08% | ~ | 7.7782 | 0.18% | ~ | 9.3937 | 0.51% | ~ | 7.7832 |
| | AMDKD-POMO | 6.1820 | 0.15% | ~ | 7.7916 | 0.35% | ~ | 9.4473 | 1.08% | ~ | 7.8070 |
| | Omni-POMO‡ | 6.2351 | 1.01% | 34s | 7.8650 | 1.29% | 2.5m | 9.4958 | 1.60% | 37s | 7.8653 |
| | Ours-inter | *6.1758* | *0.05%* | 36s | 7.7775 | 0.17% | 2m | 9.3883 | 0.45% | 33s | *7.7805* |
| | Ours-intra | 6.1758 | 0.05% | ~ | *7.7764* | *0.15%* | ~ | *9.3820* | *0.38%* | ~ | *7.7781* |
| CVRP | HGS | 11.9471 | - | (15.3h) | 15.5642 | - | (25.6h) | 19.0554 | - | (6.2h) | 15.5222 |
| | LKH3 | 11.9694 | 0.19% | (3.5d) | 15.6473 | 0.53% | (6.5d) | 19.2208 | 0.87% | (13h) | 15.6125 |
| | AMDKD-POMO* | 12.3561 | 3.42% | 56s | 15.8854 | 2.06% | 3m | 19.8395 | 4.12% | 33s | 16.0270 |
| | POMO-60 | **12.0656** | **0.99%** | ~ | 16.0914 | 3.39% | ~ | 20.2573 | 6.31% | ~ | 16.1381 |
| | POMO-100 | 12.2531 | 2.56% | ~ | **15.7544** | **1.22%** | ~ | 19.6856 | 3.31% | ~ | 15.8977 |
| | POMO-150 | 12.4322 | 4.06% | ~ | 15.8924 | 2.11% | ~ | **19.3683** | **1.64%** | ~ | 15.8976 |
| | POMO-random | 12.2758 | 2.75% | ~ | 15.7942 | 1.48% | ~ | 19.6121 | 2.92% | ~ | 15.8940 |
| | AMDKD-POMO | 12.1487 | 1.69% | ~ | 15.8119 | 1.72% | ~ | 19.5280 | 2.48% | ~ | 15.8362 |
| | Omni-POMO‡ | 12.2996 | 2.95% | 45s | 15.9878 | 2.72% | 2.5m | 19.5975 | 2.85% | 45s | 15.9616 |
| | Ours-inter | *12.0660* | *1.00%* | 56s | 15.7848 | 1.42% | 3m | 19.4109 | 1.87% | 33s | *15.7539* |
| | Ours-intra | 12.0663 | 1.00% | ~ | *15.7781* | *1.37%* | ~ | *19.3938* | *1.78%* | ~ | *15.7461* |

**Bold** and *italics* refer to the best and the second-best performance, respectively, among all deep models.
~ The inference time of a method is equal to that of the preceding method in the row above, since those deep models except for Omni-POMO utilize the original POMO architecture and result in the same inference efficiency.
‡ The training size range of Omni-POMO is [50, 200], which is broader than our [60, 150].

## 5 EXPERIMENTS

To demonstrate the effectiveness of the proposed framework, we apply it to a well-known and strong deep model, i.e., POMO (Kwon et al., 2020), and conduct comprehensive experiments on two representative routing problems, i.e., TSP and CVRP (Kool et al., 2018; Wu et al., 2021), respectively.

**Training setups.** We adhere to most of the setups in POMO. For our approach, we set the ascending problem sizes $\{N_1, N_2, ..., N_K\}$ to $\{60, 70, ..., 150\}$ with $K = 10$ and $n = 10$. Note that these sizes could be flexibly adjusted to other incremental values. We train our approach for a total of $E$ ($E = 2000$) epochs, with each size trained for $E_p$ ($E_p = 200$) epochs, ensuring robust performance across the wide range of problem sizes. Specifically, the update interval of the exemplar model is $E_{inter} = 200$ epochs for the inter-task regularization scheme and $E_{intra} = 25$ epochs for the intra-task one. The batch size is set to 64 (32 when the sizes exceed 100) for both TSP and CVRP.

**Inference setups.** Complying with the established convention (Kool et al., 2018), we randomly generate instances following the uniform distribution for both seen and unseen problem sizes during the training phase. Pertaining to the former, we select the three most representative sizes from the set of $K$ training sizes aforementioned, encompassing the minimum size of 60 (with 10,000 instances), the median size of 100 (with 10,000 instances), and the maximum size of 150 (with 1,000 instances). Pertaining to the latter, we consider three larger unseen sizes, i.e., 200, 300 and 500 (with 128 instances for each), to further assess the generalizability. We conduct all experiments including the training and evaluation on a Linux server equipped with TITAN XP GPUs (with 12 GB memory) and Intel Xeon E5-2660 CPUs at 2.0 GHz. Our dataset and code in Pytorch will be made available.

### 5.1 COMPARISON ANALYSIS

We first verify the effectiveness of our approach on seen sizes during training for both TSP and CVRP, and the results are displayed in Table 1. Specifically, we compare our approach with (1) highly specialized VRP solvers: Concorde (Applegate et al., 2020) and LKH3 (Helsgaun, 2017) for TSP, the hybrid genetic search (HGS) (Vidal, 2022) and LKH3 for CVRP; (2) learning-oriented deep models: POMO-based methods, including the original POMO (Kwon et al., 2020), AMDKD-

Table 2: Generalization results on TSP and CVRP (unseen scales).

| | Method | Test on N=200 | | | Test on N=300 | | | Test on N=500 | | | Average of Total costs |
|---|---|---|---|---|---|---|---|---|---|---|---|
| | | Obj. | Gap | Time# | Obj. | Gap | Time# | Obj. | Gap | Time# | |
| TSP | Concorde | 10.6683 | - | (8m) | 12.9534 | - | (11m) | 16.5219 | - | (17m) | 13.3812 |
| | LKH3 | 10.6683 | 0.00% | (25m) | 12.9534 | 0.00% | (47m) | 16.5219 | 0.00% | (1.2h) | 13.3812 |
| | AMDKD-POMO* | 10.9651 | 2.78% | 10s | 13.9793 | 7.92% | 33s | 19.4197 | 17.54% | 2.5m | 14.7880 |
| | POMO-60 | 11.3360 | 6.27% | ~ | 14.8162 | 14.38% | ~ | 20.5835 | 24.58% | ~ | 15.5786 |
| | POMO-100 | 10.8464 | 1.67% | ~ | 13.8730 | 7.10% | ~ | 20.1985 | 22.25% | ~ | 14.9726 |
| | POMO-150 | 10.7752 | 1.00% | ~ | *13.2922* | *2.62%* | ~ | 18.0793 | 9.43% | ~ | 14.0489 |
| | POMO-random | 10.8397 | 1.61% | ~ | 13.8212 | 6.70% | ~ | 19.0881 | 15.53% | ~ | 14.5830 |
| | AMDKD-POMO | 10.9054 | 2.22% | ~ | 13.4472 | 3.81% | ~ | 18.4477 | 11.66% | ~ | 14.2668 |
| | Omni-POMO‡ | 10.8923 | 2.10% | 11s | 13.4044 | 3.48% | 33s | **17.8146** | **7.82%** | 2.6m | *14.0371* |
| | Ours-inter | *10.7631* | *0.89%* | 10s | 13.2942 | 2.63% | 33s | 18.1047 | 9.58% | 2.5m | 14.0540 |
| | Ours-intra | **10.7444** | **0.71%** | ~ | **13.2263** | **2.11%** | ~ | *18.0267* | *9.11%* | ~ | **13.9992** |
| CVRP | HGS | 21.9737 | - | (1.1h) | 25.8417 | - | (1.6h) | 31.0308 | - | (2.5h) | 26.6514 |
| | LKH3 | 22.2146 | 1.10% | (2.4h) | 26.2184 | 1.46% | (3.2h) | 31.5213 | 1.58% | (5.3h) | 26.6514 |
| | AMDKD-POMO* | 23.8507 | 8.54% | 12s | 30.7218 | 17.18% | 38s | 48.1260 | 52.68% | 3m | 34.2328 |
| | POMO-60 | 24.0638 | 9.51% | ~ | 29.6416 | 14.71% | ~ | 38.8480 | 25.19% | ~ | 30.8511 |
| | POMO-100 | 23.2783 | 5.94% | ~ | 28.9372 | 11.98% | ~ | 37.9132 | 22.18% | ~ | 30.0429 |
| | POMO-150 | *22.4706* | *2.26%* | ~ | 26.8810 | 4.02% | ~ | 33.7746 | 8.84% | ~ | 27.7087 |
| | POMO-random | 23.2016 | 5.59% | ~ | 28.1393 | 8.89% | ~ | 35.6822 | 14.99% | ~ | 29.0077 |
| | AMDKD-POMO | 22.7842 | 3.69% | ~ | 27.4462 | 4.68% | ~ | 34.0650 | 9.78% | ~ | 28.0985 |
| | Omni-POMO‡ | 22.6562 | 3.11% | 13s | 26.8707 | 3.98% | 38s | **33.1435** | **6.81%** | 4m | *27.5568* |
| | Ours-inter | 22.4847 | 2.33% | 12s | *26.8134* | *3.76%* | 38s | 33.6337 | 8.39% | 3m | 27.6439 |
| | Ours-intra | **22.4436** | **2.14%** | ~ | **26.6884** | **3.28%** | ~ | *33.4200* | *7.70%* | ~ | **27.5173** |

POMO (Bi et al., 2022) and Omni-POMO (Zhou et al., 2023) for both TSP and CVRP. For POMO, we retrain the model on each problem size with an equivalent number of epochs as our approach for a fair comparison, e.g., *POMO-60* signifying the model meticulously trained on size 60. AMDKD-POMO improved the cross-distribution generalization of POMO using knowledge distillation, where we retrain it following our training setups by tailoring teacher models to align with our exemplar sizes. Besides, we also show the results of its open-sourced pretrained models on the largest available sizes, i.e., AMDKD-POMO*. Furthermore, Omni-POMO is a recent meta-learning framework to improve generalization across size and distribution of POMO for VRPs, where we report their results based on the direct use of their open-sourced pretrained models. Regarding our approach, two distinct variations with inter-task and intra-task regularization schemes are denoted as *Ours-inter* and *Ours-intra*, respectively. Every method is assessed using the data augmentation of POMO, and the results without augmentation are reported in Appendix A. The total inference time is also reported for all methods, i.e., GPU time for deep models and CPU time for traditional solvers.

From Table 1, we observe that Ours-inter performs slightly superior to Ours-intra on small sizes (e.g., 60), but the other way round on larger sizes (e.g., 100 and 150) for both TSP and CVRP. This is reasonable since intra-task regularization concentrate more on efficiently learning the latest larger sizes. While specially designed to improve the cross-distribution generalization of POMO, AMDKD-POMO* still suffers from the cross-size generalization issue. Moreover, in comparison with the original POMO trained on a specific size, both Ours-inter and Ours-intra achieve competitive performance concerning the objective values on that size for both TSP and CVRP, while significantly outperforming those POMO models concerning average objective values over the three sizes (refer to the final column). Furthermore, both Ours-inter and Ours-intra outstrip POMO-random, AMDKD-POMO and Omni-POMO across all sizes for both TSP and CVRP with comparable inference time, even if Omni-POMO utilizes training instances with larger upper sizes (i.e., 200).

## 5.2 GENERALIZATION ANALYSIS

We further evaluate all methods on unseen larger sizes and gathered the results in Table 2. As revealed, the cross-size generalization issue of AMDKD-POMO* is more pronounced, leading to a substantial deterioration in performance. Ours-inter surpasses POMO-random, POMO-60, POMO-100 and AMDKD-POMO in terms of average objective values over the three sizes, and achieves competitive performance to POMO-150 for both TSP and CVRP. Leveraging intra-task regularization to prioritize the learning of the latest larger sizes, Ours-intra further outperforms POMO-150 across all sizes. Moreover, both Ours-inter and Ours-intra deliver superior performance to Omni-

Table 3: Generalization performance on instances ($50 \le N \le 500$) from benchmark instances.

| | POMO-60 | POMO-100 | POMO-150 | AMDKD-POMO | Omni-POMO | Ours-Inter |
|---|---|---|---|---|---|---|
| TSPLIB | 9.71% | 4.49% | 4.18% | 5.17% | 3.11% | 4.07% |
| CVRPLIB | 13.59% | 12.30% | 9.21% | 7.09% | 5.83% | 5.45% |

POMO on sizes 200 and 300 for both TSP and CVRP. Although our approach exhibits slightly inferior performance to Omni-POMO on size 500, it is worth noting that Omni-POMO is trained on a broader range of sizes (including larger ones up to 200), which inherently offers Omni-POMO the potential for superior performance on larger sizes. Furthermore, Ours-intra consistently achieves lower average objective values than Omni-POMO over the three sizes for both TSP and CVRP, which further underscores the effectiveness of our approach.

In Table 3, we extend the evaluation to well-established benchmark datasets TSPLIB (Reinelt, 1991) and CVRPLIB (Uchoa et al., 2017) with size smaller than 500, and report the average gaps for all methods. AMDKD-POMO* is omitted due to its clear inferiority as demonstrated in Table 1 and Table 2. Notably, benchmark datasets usually encompass a diverse range of sizes and customer distribution patterns. For our approach, we emphasize Ours-intra given its decent performance demonstrated earlier. The results indicate that our approach outstrips AMDKD-POMO and POMO trained on three specific sizes when evaluated on both benchmark datasets, which showcases that our approach could effectively enhance the the cross-size generalization ability of a backbone model. Moreover, despite the fact that Omni-POMO is specially designed to enhance generalization across both size and distribution, our approach still delivers desirable performance that is competitive to Omni-POMO. This further showcases the efficiency and effectiveness of our approach (see Appendix B for details).

## 5.3 ABLATION STUDY

In Table 8, we conduct an ablation study to clarify the effectiveness of each component of our approach on TSP and CVRP (refer to Appendix C.1), where only one regularization scheme can be used in our approach to keep a stable update of the exemplar model. The markers "✓" and "×" denote the utilization or exclusion of the

Table 4: Ablation study on TSP.

| ER | Inter-task | Intra-task | N=60 | | N=100 | | N=150 | |
|---|---|---|---|---|---|---|---|---|
| | | | Obj. | Gap | Obj. | Gap | Obj. | Gap |
| × | × | × | 6.1886 | 0.25% | 7.7898 | 0.32% | 9.3974 | 0.55% |
| × | ✓ | × | 6.1805 | 0.12% | 7.7831 | 0.24% | 9.3938 | 0.51% |
| × | × | ✓ | 6.1809 | 0.13% | 7.7829 | 0.24% | 9.3885 | 0.45% |
| ✓ | × | × | 6.1789 | 0.10% | 7.7860 | 0.28% | 9.3932 | 0.50% |
| ✓ | ✓ | × | 6.1758 | 0.05% | 7.7775 | 0.17% | 9.3883 | 0.45% |
| ✓ | × | ✓ | 6.1758 | 0.05% | 7.7764 | 0.15% | 9.3820 | 0.38% |

corresponding component, respectively. The gaps are calculated based on the solutions acquired by Concorde in Table 1. As exhibited, experience replay, inter-task and intra-task regularization schemes contribute to the reduction of objective values and optimality gaps across all sizes, affirming their effectiveness in enhancing cross-size generalizability. Further combining them together, both Ours-inter and Ours-intra (last two rows) achieve better performance. We further conduct an ablation study to demonstrate the effectiveness of the proposed regularization schemes, including both the inter-task regularization scheme and the intra-task one (see Appendix C.2 for details).

We further conduct experiments to show that our approach can consistently improve the performance across diverse sizes as the training progresses. To this end, we use the obtained models after training on each size to evaluate on both seen and unseen sizes. We present the details in Appendix D.

## 6 CONCLUSIONS AND FUTURE WORK

This paper presents a continual learning based framework to foster the cross-size generalization of deep models for VRPs. We leverage either inter-task or intra-task regularization scheme to retain the valuable insights derived from previously trained exemplar models for facilitating subsequent training. To mitigate the catastrophic forgetting, we exploit the experience replay to revisit instances of formerly trained smaller sizes. Results show that our approach not only significantly strengthens the cross-size generalization performance, but also delivers predominantly superior performance to state-of-the-art deep models specialized for the generalizability enhancement.

For future work, we will investigate automatically selecting the inter-task or intra-task regularization scheme to further enhance our approach. Scaling up to substantially large problem instances is also important. Bolstered by the superior cross-size generalization ability, we will further improve the CL framework to train reliable deep models for handling large-scale VRPs, e.g., in a divide-and-conquer manner. Additionally, explicitly enhancing the cross-distribution generalization in the proposed CL framework could further unleash the potential of our approach in real-world applications.

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

## A FULL RESULTS ON TSP AND CVRP (SEEN SCALES)

In Table 1, regarding the learning-oriented methods, we mainly displayed their results yielded with the data augmentation strategy (A). Here we show the full results with (A) and without (N) the data augmentation in Table 5. As shown, the inference time without data augmentation is significantly shorter than that with data augmentation. On the other hand, it is clear that both Ours-inter and Ours-intra achieve the best average performance among all deep models, no matter the data augmentation strategy is used or not.

Table 5: Comparison results on TSP and CVRP (seen scales).

| | Method | Test on N=60 | | | Test on N=100 | | | Test on N=150 | | | Average of |
| | | Obj. | Gap | Time# | Obj. | Gap | Time# | Obj. | Gap | Time# | Total costs |
|---|---|---|---|---|---|---|---|---|---|---|---|
| TSP | Concorde | 6.1729 | - | (7m) | 7.7646 | - | (1.7h) | 9.3462 | - | (22m) | 7.7612 |
| | LKH3 | 6.1729 | 0.00% | (14m) | 7.7646 | 0.00% | (9.8h) | 9.3462 | 0.00% | (2.1h) | 7.7612 |
| | AMDKD-POMO* | 6.1828 | 0.16% | 36s | 7.7930 | 0.37% | 2m | 9.4539 | 1.15% | 33s | 7.8092 |
| | POMO-60 | **6.1746** | **0.03%** | ~ | 7.8050 | 0.52% | ~ | 9.5909 | 2.62% | ~ | 7.8568 |
| | POMO-100 | 6.1768 | 0.06% | ~ | **7.7753** | **0.14%** | ~ | 9.3987 | 0.56% | ~ | 7.7836 |
| | POMO-150 | 6.1928 | 0.32% | ~ | 7.7875 | 0.30% | ~ | **9.3812** | **0.36%** | ~ | 7.7868 |
| | POMO-random | 6.1778 | 0.08% | ~ | 7.7782 | 0.18% | ~ | 9.3937 | 0.51% | ~ | 7.7832 |
| | AMDKD-POMO | 6.1820 | 0.15% | ~ | 7.7916 | 0.35% | ~ | 9.4473 | 1.08% | ~ | 7.8070 |
| | Omni-POMO‡ | 6.2351 | 1.01% | 34s | 7.8650 | 1.29% | 2.5m | 9.4958 | 1.60% | 37s | 7.8653 |
| | Ours-inter | *6.1758* | *0.05%* | 36s | 7.7775 | 0.17% | 2m | 9.3883 | 0.45% | 33s | *7.7805* |
| | Ours-intra | 6.1758 | 0.05% | ~ | *7.7764* | *0.15%* | ~ | *9.3820* | *0.38%* | ~ | **7.7781** |
| CVRP | HGS | 11.9471 | - | (15.3h) | 15.5642 | - | (25.6h) | 19.0554 | - | (6.2h) | 15.5222 |
| | LKH3 | 11.9694 | 0.19% | (3.5d) | 15.6473 | 0.53% | (6.5d) | 19.2208 | 0.87% | (13h) | 15.6125 |
| | AMDKD-POMO* | 12.3561 | 3.42% | 56s | 15.8854 | 2.06% | 3m | 19.8395 | 4.12% | 33s | 16.0270 |
| | POMO-60 | **12.0656** | **0.99%** | ~ | 16.0914 | 3.39% | ~ | 20.2573 | 6.31% | ~ | 16.1381 |
| | POMO-100 | 12.2531 | 2.56% | ~ | **15.7544** | **1.22%** | ~ | 19.6856 | 3.31% | ~ | 15.8977 |
| | POMO-150 | 12.4322 | 4.06% | ~ | 15.8924 | 2.11% | ~ | **19.3683** | **1.64%** | ~ | 15.8976 |
| | POMO-random | 12.2758 | 2.75% | ~ | 15.7942 | 1.48% | ~ | 19.6121 | 2.92% | ~ | 15.8940 |
| | AMDKD-POMO | 12.1487 | 1.69% | ~ | 15.8119 | 1.72% | ~ | 19.5280 | 2.48% | ~ | 15.8362 |
| | Omni-POMO‡ | 12.2996 | 2.95% | 45s | 15.9878 | 2.72% | 2.5m | 19.5975 | 2.85% | 45s | 15.9616 |
| | Ours-inter | *12.0660* | *1.00%* | 56s | 15.7848 | 1.42% | 3m | 19.4109 | 1.87% | 33s | *15.7539* |
| | Ours-intra | 12.0663 | 1.00% | ~ | *15.7781* | *1.37%* | ~ | *19.3938* | *1.78%* | ~ | **15.7461** |

**Bold** and *italics* refer to the best and the second-best performance, respectively, among all deep models.
~ The inference time of a method is equal to that of the preceding method in the row above, since those deep models except for Omni-POMO utilize the original POMO architecture and result in the same inference efficiency.
‡ The training size range of Omni-POMO is [50, 200], which is broader than our [60, 150].

## B DETAILED GENERALIZATION RESULTS ON BENCHMARK DATASETS.

We evaluate all methods on the classic benchmark datasets, including TSPLIB (Reinelt, 1991) and CVRPLIB (Uchoa et al., 2017), in which we choose representative instances with size $N \in [50, 500]$. Note that both benchmark datasets encompass a diverse range of sizes and distributions. The detailed results are shown in Table 6 and Table 7, where the gaps are calculated based on the optimal solution values for the instances. From two tables, we observe that our approach outperforms AMDKD-POMO and all POMO models trained on specific sizes on both benchmark datasets. Note that Omni-POMO is explicitly specialized for improving generalization across both size and distribution, which has the potential to achieve desirable performance on benchmark datasets whose instances encompass a diverse range of sizes and distributions. More specific, the setting that explicitly training with more diverse sizes and distributions, will inherently favor Omni-POMO over our approach. Whereas, our approach still yields competitive performance in comparison against Omni-POMO, e.g., with the average gap of 5.45% (Ours) VS. 5.83% (Omni-POMO) on CVRPLIB, which further underscores the effectiveness of our approach.

Table 6: Detailed generalization results on instances from TSPLIB.

| Instance | Opt. | POMO-60 Obj. | POMO-60 Gap | POMO-100 Obj. | POMO-100 Gap | POMO-150 Obj. | POMO-150 Gap | AMDKD-POMO Obj. | AMDKD-POMO Gap | Omni-POMO[‡] Obj. | Omni-POMO[‡] Gap | Ours Obj. | Ours Gap |
|---|---|---|---|---|---|---|---|---|---|---|---|---|---|
| berlin52 | 7544 | 7544 | 0.00% | 7545 | 0.01% | 7613 | 0.92% | 7545 | 0.01% | 8003 | 6.08% | 7544 | 0.00% |
| st70 | 675 | 677 | 0.30% | 677 | 0.30% | 678 | 0.44% | 677 | 0.30% | 680 | 0.74% | 677 | 0.30% |
| eil76 | 538 | 547 | 1.67% | 544 | 1.12% | 549 | 2.05% | 550 | 3.77% | 557 | 3.53% | 544 | 1.12% |
| rd100 | 7910 | 7920 | 0.13% | 7910 | 0.00% | 7952 | 0.53% | 7934 | 0.30% | 7958 | 0.61% | 7910 | 0.00% |
| KroA100 | 21282 | 21786 | 2.34% | 21667 | 1.81% | 21596 | 1.48% | 22077 | 3.74% | 21305 | 0.11% | 21738 | 2.14% |
| KroB100 | 22141 | 22941 | 3.61% | 22370 | 1.03% | 22575 | 1.96% | 22745 | 2.73% | 22650 | 2.30% | 22640 | 2.25% |
| lin105 | 14379 | 14808 | 2.98% | 14557 | 1.24% | 14808 | 2.98% | 14898 | 3.61% | 14819 | 3.06% | 14753 | 2.60% |
| pr124 | 59030 | 59031 | 0.00% | 59388 | 0.61% | 59595 | 0.96% | 59521 | 0.83% | 59238 | 0.35% | 59164 | 0.23% |
| ch130 | 6110 | 6188 | 1.28% | 6133 | 0.38% | 6142 | 0.52% | 6159 | 0.80% | 6251 | 2.31% | 6119 | 0.15% |
| pr136 | 96772 | 100459 | 38.12% | 97540 | 0.80% | 97668 | 0.93% | 97951 | 1.22% | 97780 | 1.04% | 97258 | 0.50% |
| gr137 | 699 | 746 | 6.72% | 755 | 8.01% | 759 | 8.58% | 773 | 10.59% | 772 | 10.44% | 747 | 6.87% |
| ch150 | 6528 | 6679 | 2.31% | 6559 | 0.48% | 6579 | 0.78% | 6583 | 0.82% | 6586 | 0.89% | 6559 | 0.48% |
| KroA200 | 29368 | 31819 | 8.35% | 30415 | 3.57% | 30015 | 2.20% | 30672 | 4.44% | 29823 | 1.55% | 29951 | 1.99% |
| KroB200 | 29437 | 32020 | 8.78% | 30880 | 4.90% | 30172 | 2.50% | 30990 | 5.28% | 29814 | 1.28% | 30792 | 4.60% |
| ts225 | 126643 | 135704 | 7.16% | 130990 | 3.43% | 128045 | 1.14% | 128911 | 1.79% | 128770 | 1.68% | 129297 | 2.10% |
| a280 | 2579 | 3101 | 20.24% | 2951 | 0.45% | 2788 | 8.10% | 2809 | 8.92% | 2695 | 4.50% | 2828 | 9.65% |
| rd400 | 15281 | 18055 | 18.15% | 17342 | 13.49% | 16155 | 5.72% | 16160 | 5.75% | 15948 | 4.37% | 15968 | 4.50% |
| fl417 | 11861 | 14319 | 20.72% | 14396 | 21.37% | 14225 | 19.93% | 14004 | 18.07% | 12683 | 6.93% | 13932 | 17.46% |
| pcb442 | 50778 | 71914 | 41.62% | 62145 | 22.39% | 54683 | 7.69% | 63643 | 25.34% | 59761 | 17.69% | 61152 | 20.43% |
| Avg. Gap | 0.00% | - | 9.71% | - | 4.49% | - | 4.18% | - | 5.17% | - | 3.11% | - | 4.07% |

Table 7: Detailed generalization results on instances from CVRPLIB.

| Instance | Opt. | POMO-60 Obj. | POMO-60 Gap | POMO-100 Obj. | POMO-100 Gap | POMO-150 Obj. | POMO-150 Gap | AMDKD-POMO Obj. | AMDKD-POMO Gap | Omni-POMO[‡] Obj. | Omni-POMO[‡] Gap | Ours Obj. | Ours Gap |
|---|---|---|---|---|---|---|---|---|---|---|---|---|---|
| A-n53-k7 | 1010 | 1315 | 30.20% | 1318 | 30.50% | 1152 | 14.06% | 1111 | 10.00% | 1105 | 9.41% | 1136 | 12.48% |
| A-n60-k9 | 1354 | 1741 | 28.58% | 1739 | 28.43% | 1657 | 22.38% | 1574 | 16.25% | 1465 | 8.20% | 1453 | 7.31% |
| A-n80-k10 | 1763 | 2692 | 52.69% | 2740 | 55.42% | 2816 | 59.73% | 2136 | 21.16% | 2127 | 20.65% | 2100 | 19.12% |
| X-n101-k25 | 27591 | 29786 | 7.96% | 29287 | 6.15% | 29398 | 6.55% | 29306 | 6.22% | 29442 | 6.71% | 28533 | 3.41% |
| X-n110-k13 | 14971 | 15530 | 3.73% | 15161 | 1.27% | 15130 | 1.06% | 15202 | 1.54% | 15285 | 2.10% | 15102 | 0.88% |
| X-n120-k6 | 13332 | 14239 | 6.80% | 14570 | 9.29% | 14060 | 5.46% | 14010 | 5.09% | 13944 | 4.59% | 13882 | 4.13% |
| X-n129-k18 | 28940 | 30154 | 4.20% | 29569 | 2.17% | 29343 | 1.39% | 29702 | 2.63% | 29975 | 3.58% | 29306 | 1.27% |
| X-n139-k10 | 13590 | 14269 | 5.00% | 14080 | 3.61% | 13855 | 1.95% | 13890 | 2.21% | 14019 | 3.16% | 13812 | 1.63% |
| X-n148-k46 | 43448 | 46146 | 6.21% | 47621 | 9.61% | 45850 | 5.53% | 46451 | 6.91% | 46438 | 6.88% | 45600 | 4.95% |
| X-n157-k13 | 16876 | 17663 | 4.66% | 18302 | 8.45% | 18340 | 8.68% | 17523 | 3.83% | 17107 | 1.37% | 17414 | 3.19% |
| X-n167-k10 | 20557 | 22306 | 8.51% | 21297 | 3.60% | 20995 | 2.13% | 21068 | 2.49% | 21436 | 4.28% | 20960 | 1.96% |
| X-n190-k8 | 16980 | 18757 | 10.47% | 18164 | 6.97% | 18140 | 6.83% | 18169 | 7.00% | 17645 | 3.92% | 17770 | 4.65% |
| X-n200-k36 | 58578 | 62737 | 7.10% | 61933 | 5.73% | 61397 | 4.81% | 61384 | 4.79% | 61496 | 4.98% | 61514 | 5.01% |
| X-n251-k28 | 38684 | 42430 | 9.69% | 41360 | 6.92% | 40024 | 3.47% | 40595 | 4.94% | 40059 | 3.55% | 40046 | 3.52% |
| X-n298-k31 | 34231 | 38749 | 13.20% | 38611 | 12.80% | 35663 | 4.18% | 37221 | 8.74% | 36384 | 6.29% | 35779 | 4.52% |
| X-n351-k40 | 25896 | 30289 | 16.96% | 28343 | 9.45% | 27952 | 7.94% | 28315 | 9.34% | 27515 | 6.25% | 27487 | 6.14% |
| X-n401-k29 | 66154 | 72209 | 9.15% | 71173 | 7.59% | 69641 | 5.27% | 69227 | 4.65% | 68234 | 3.14% | 69682 | 5.33% |
| X-n449-k29 | 55233 | 63569 | 15.09% | 61915 | 12.08% | 58418 | 5.77% | 59929 | 8.50% | 58037 | 5.08% | 58563 | 6.03% |
| X-n491-k59 | 66483 | 78463 | 18.02% | 75620 | 13.74% | 71668 | 7.80% | 72087 | 8.43% | 70923 | 6.68% | 71787 | 7.98% |
| Avg. Gap | 0.00% | - | 13.59% | - | 12.30% | - | 9.21% | - | 7.09% | - | 5.83% | - | 5.45% |

# C  MORE ABLATION STUDY RESULTS

## C.1  ABLATION STUDY ON COMPONENTS OF OUR APPROACH FOR CVRP

In Table 4, we display the ablation study on the effectiveness of each component of our approach for TSP. Here we show the corresponding results for CVRP in Table 8. As revealed, experience replay, inter-task and intra-task regularization schemes contribute to the reduction of objective values and optimality gaps across all sizes, affirming their effectiveness in enhancing cross-size generalizability. Further combining experience replay and regularization scheme together, both Ours-inter and Ours-intra (last two rows) achieve better performance.

Table 8: Ablation study on CVRP.

| ER | Inter-task | Intra-task | N=60 Obj. | N=60 Gap | N=100 Obj. | N=100 Gap | N=150 Obj. | N=150 Gap |
|---|---|---|---|---|---|---|---|---|
| × | × | × | 12.1174 | 1.43% | 15.8528 | 1.86% | 19.5024 | 2.35% |
| × | ✓ | × | 12.0829 | 1.14% | 15.8109 | 1.59% | 19.4472 | 2.06% |
| × | × | ✓ | 12.0855 | 1.16% | 15.8085 | 1.57% | 19.4289 | 1.96% |
| ✓ | × | × | 12.0786 | 1.10% | 15.8187 | 1.64% | 19.4389 | 2.01% |
| ✓ | ✓ | × | 12.0660 | 1.00% | 15.7848 | 1.42% | 19.4109 | 1.87% |
| ✓ | × | ✓ | 12.0663 | 1.00% | 15.7781 | 1.37% | 19.3938 | 1.78% |

## C.2 ABLATION STUDY ON REGULARIZATION SCHEMES

To verify the effectiveness of the regularization schemes designed in our approach, we further analyze the evaluation results of our accomplished models after training on each size with inter-task regularization scheme, intra-task regularization scheme, and without regularization scheme, across three sizes (i.e., 60, 100 and 150) for both TSP and CVRP. The corresponding curves are depicted in Figure 3 and Figure 4, respectively, where we also present the average objective values across those sizes, in order to show the overall improved cross-size generalization performance. As revealed, both inter-task and intra-task regularization schemes accelerate the learning efficiency and boost the overall results in comparison with the one without regularization scheme for both TSP and CVRP, which showcases the effectiveness of the proposed regularization schemes. Moreover, inter-task regularization scheme is superior to the intra-task one in terms of the learning efficiency on smaller sizes (i.e., 60), while the other way round on larger sizes (i.e., 100 and 150). This is reasonable since the intra-task regularization scheme concentrate more on efficiently learning the latest lager sizes. Furthermore, the intra-task achieves the fastest learning efficiency and the lowest objective value when averaging the objective values across all three sizes.

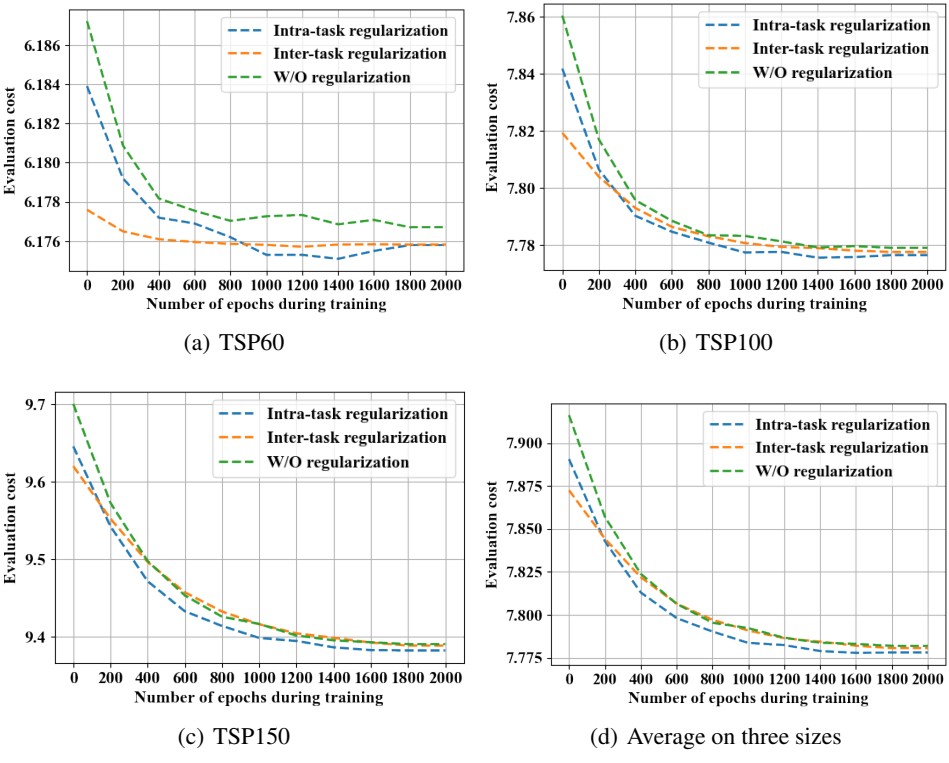

(a) TSP60      (b) TSP100

(c) TSP150      (d) Average on three sizes

Figure 3: Curves of learning progress for our approach on TSP.

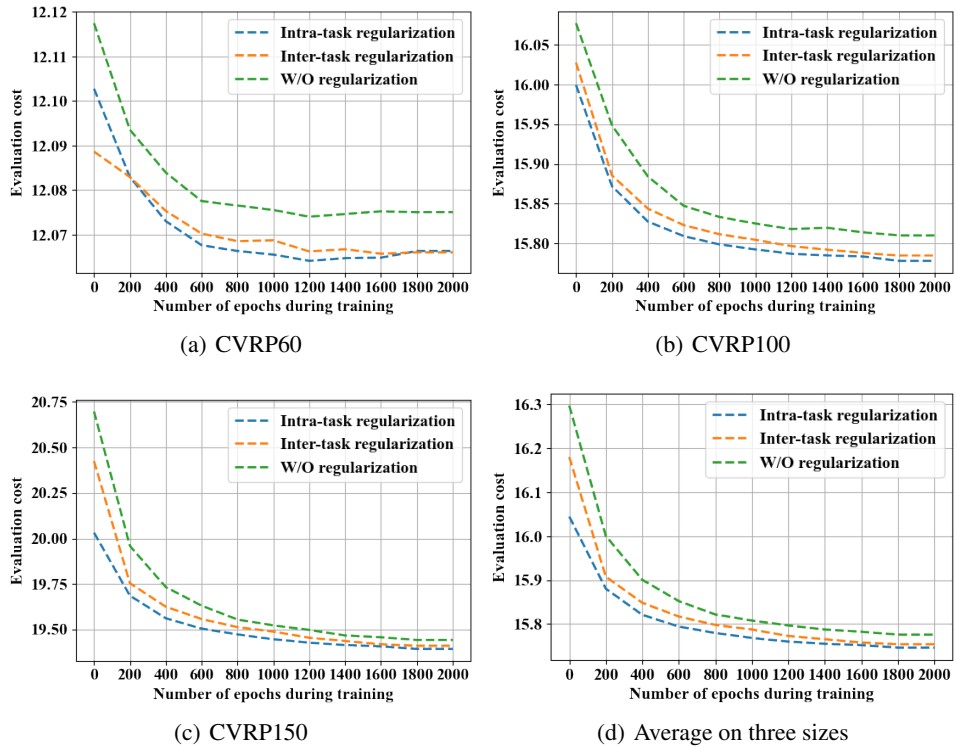

Figure 4: Curves of learning progress for our approach on CVRP.

Table 9: Validation results on seen and unseen sizes.

|  | Method | Test on N=100 | | Test on N=150 | | Test on N=200 | |
|---|---|---|---|---|---|---|---|
|  |  | Obj. | Gap | Obj. | Gap | Obj. | Gap |
| TSP | Ours (after training on 60) | 7.8419 | 1.00% | 9.6460 | 3.21% | 11.2975 | 5.90% |
|  | Ours (after training on 70) | 7.8065 | 0.54% | 9.5426 | 2.10% | 11.1413 | 4.43% |
|  | Ours (after training on 80) | 7.7902 | 0.33% | 9.4715 | 1.34% | 11.0082 | 3.19% |
|  | Ours (after training on 90) | 7.7846 | 0.26% | 9.4327 | 0.93% | 10.9116 | 2.28% |
|  | Ours (after training on 100) | 7.7808 | 0.21% | 9.4137 | 0.72% | 10.8585 | 1.78% |
|  | Ours (after training on 110) | 7.7773 | 0.16% | 9.3982 | 0.56% | 10.8029 | 1.26% |
|  | Ours (after training on 120) | 7.7775 | 0.17% | 9.3943 | 0.52% | 10.7880 | 1.12% |
|  | Ours (after training on 130) | 7.7755 | 0.14% | 9.3861 | 0.43% | 10.7629 | 0.89% |
|  | Ours (after training on 140) | 7.7757 | 0.14% | 9.3824 | 0.39% | 10.7526 | 0.79% |
|  | Ours (after training on 150) | 7.7764 | 0.15% | 9.3820 | 0.38% | 10.7445 | 0.71% |
| CVRP | Ours (after training on 60) | 15.9992 | 2.23% | 20.0320 | 4.22% | 23.5642 | 6.08% |
|  | Ours (after training on 70) | 15.8721 | 1.42% | 19.6862 | 2.42% | 23.0514 | 3.77% |
|  | Ours (after training on 80) | 15.8275 | 1.13% | 19.5619 | 1.78% | 22.8025 | 2.65% |
|  | Ours (after training on 90) | 15.8092 | 1.02% | 19.5051 | 1.48% | 22.7007 | 2.19% |
|  | Ours (after training on 100) | 15.7987 | 0.95% | 19.4732 | 1.31% | 22.6235 | 1.84% |
|  | Ours (after training on 110) | 15.7924 | 0.91% | 19.4466 | 1.18% | 22.5745 | 1.62% |
|  | Ours (after training on 120) | 15.7870 | 0.88% | 19.4282 | 1.08% | 22.5361 | 1.45% |
|  | Ours (after training on 130) | 15.7849 | 0.86% | 19.4148 | 1.01% | 22.4948 | 1.26% |
|  | Ours (after training on 140) | 15.7835 | 0.85% | 19.4075 | 0.97% | 22.4695 | 1.15% |
|  | Ours (after training on 150) | 15.7781 | 0.82% | 19.3938 | 0.90% | 22.4436 | 1.03% |

## D    VALIDATION ON BOTH SEEN AND UNSEEN SIZES

We evaluate the obtained models after training on each size with intra-task regularization scheme. On both seen and unseen sizes, i.e., 100, 150 and 200, these models are evaluated to showcase that our approach can consistently improve the performance of the deep model, as the training progresses. The results are summarized in Table 9, where the gaps are calculated based on the solutions

acquired by Concorde for TSP and HGS for CVRP in Table 1. We observe that when we evaluate the models on size 100, it progressively reduces the gaps before the training process reaching size 100. Furthermore, it effectively retains the acquired knowledge and superiority on size 100, after further trained on instances with larger sizes. The models trained after size 100 still show decreasing gaps on size 100. Similarly, in the case of testing on the larger sizes, i.e., 150 and 200, our approach consistently enhances the performance during the whole training phase. These findings highlight that our approach excels not only in preserving superior performance on small sizes but also in consistently enhancing performance on larger sizes.

## E   DETAILS OF POMO MODEL.

POMO uses the encoder-decoder structure to sequentially construct solutions for VRPs. Specifically, the encoder projects problem-specific features into high-dimensional node embeddings for informative representation learning. Afterwards, the decoder sequentially constructs a solution $\tau = \{\tau_1, \tau_2, ..., \tau_{T_c}\}$ with $T_c$ steps for a VRP instance of problem size $N$, conditioned on the node embeddings and partial tour at each step. Specifically, $T_c = N$ for TSP, and $T_c \geq N$ for CVRP as the depot node could be visited multiple times.

### E.1   THE ENCODER

The encoder first embeds problem-specific features to higher-dimensional space, then passes them to stacked attention layers to extract useful information for better representation. The problem-specific features of node $v_i, i \in \{1, 2, ..., N\}$ contain 2-dimensional location coordinates (for both TSP and CVRP) and 1-dimensional demand vector (for CVRP only), which are linearly projected to initial node embedding $h_i^0$ of 128-dimension [1]. Then they are processed through $L = 6$ attention layers with different parameters to derive the final node embedding $h_i^L$, where each attention layer is composed of a multi-head attention (MHA) sub-layer and a feed-forward (FF) sub-layer. Following the original design of the Transformer architecture [2], both the outputs of the MHA sub-layer and the FF sub-layer are followed by a skip-connection layer [3] and a batch normalization (BN) layer [4] as below,

$$\tilde{h}_i = \text{BN}(h_i^l + \text{MHA}(h_i^l)), \tag{6}$$

$$h_i^{l+1} = \text{BN}(\tilde{h}_i + \text{FF}(\tilde{h}_i)). \tag{7}$$

**MHA sub-layer.** The MHA sub-layer employs a multi-head self-attention mechanism [2] with $M = 8$ heads to compute the attention weights between each two nodes. Specifically, the *query/key/value* proposed in [2] are defined with $d_k = d/M$ dimension as below,

$$q_i^{l,m} = W_Q^{l,m} h_i^l, \ k_i^{l,m} = W_K^{l,m} h_i^l, \ v_i^{l,m} = W_V^{l,m} h_i^l. \tag{8}$$

Then the attention weights are computed by using the Softmax activation function to represent the correlation between each two nodes as follows,

$$u_{ij}^{l,m} = \text{Softmax}\left(\frac{(q_i^{l,m})^\top (k_j^{l,m})}{\sqrt{d_k}}\right). \tag{9}$$

Finally, the $l$-th MHA sub-layer first computes the new context vectors by performing an element-wise multiplication of the attention weights with *value*, and then aggregates the information from $M$ heads as follows,

$$h_i^{l,m} = \sum_j u_{ij}^{l,m} v_j^{l,m}, \ m = 1, 2, ..., M, \tag{10}$$

$$\text{MHA}(h_i^l) = [h_i^{l,1}; h_i^{l,2}; ...; h_i^{l,M}] W_O^l, \tag{11}$$

where $W_Q^{l,m}, W_K^{l,m}, W_V^{l,m} \in \mathbb{R}^{d \times d_k}$, $W_O^l \in \mathbb{R}^{md_k \times d}$ are learnable parameters, and $[;]$ denotes the concatenate operator.

**FF sub-layer.** The FF sub-layer processes the node embeddings $\tilde{h}_i, i \in \{1, 2, ..., N\}$ through a hidden sub-layer with dimension 512 and a ReLU activation function, as follows,

$$\text{FF}(\tilde{h}_i) = W_F^1 \, \text{ReLU}(W_F^0 \tilde{h}_i + b_F^0) + b_F^1, \tag{12}$$

where $W_F^0, W_F^1, b_F^0, b_F^1$ are trainable parameters. At the end, the encoder outputs a set of node embeddings in the $L$-th layer $h_i^L, i \in \{1, 2, ..., N\}$. These embeddings will be preserved as a part of the input to the decoder for route construction.

### E.2 THE DECODER

Taking TSP as an example, the decoder first calculates the mean of node embeddings to derive the global graph embedding, i.e., $\bar{h} = \frac{1}{N} \sum_i^N h_i^L$, then defines a context vector as the combination of the graph embedding, the embeddings of the first node of the route and the last visited node. For step $t_c = 1$, we use learned parameters $v^1$ and $v^2$ as input placeholders. Note that POMO constructs $N$ solutions by taking each of the $N$ nodes as the first node to visit (i.e., $h_{\tau_1}^i = h_i$). Particularly, it defines $N$ context vectors $h_i^c, \forall i \in \{1, 2, ..., N\}$ as follows,

$$h_i^c = \begin{cases} (\bar{h}, h_i, h_{\tau_{t_c-1}}^i), & \text{if } t_c > 1 \\ (\bar{h}, v^1, v^2), & \text{if } t_c = 1 \end{cases} \tag{13}$$

where $h_{\tau_{t_c}}^i$ is the node embedding of the node visited at decoding step $t_c$ for the $i$th solution. The context vectors are then processed by a MHA layer as introduced above to generate $N$ glimpse vectors $h_i^g, i \in \{1, 2, ..., N\}$ in parallel,

$$h_i^g = \text{MHA}(W_Q^g h_i^c, \ W_K^g h, \ W_V^g h), \ i \in \{1, 2, ..., N\}, \tag{14}$$

after which the decoder computes the compatibility between the enhanced glimpses and node embeddings. Then it further calculates the probabilities of selecting the next node to visit for $N$ solutions in parallel at decoding step $t_c$ as follows,

$$c_i^{t_c} = G \cdot \tanh\left(\frac{(h_i^g W_Q)^T (h W_K)}{\sqrt{d_k}}\right), \tag{15}$$

$$p_i^{t_c} = \text{Softmax}(c_i^{t_c}), i \in \{1, 2, ..., N\}, \tag{16}$$

where $W_Q^g, W_K^g, W_V^g, W_Q, W_K$ are learnable parameter matrices, and $G$ is often set to 10 to control the entropy of $c_i^{t_c}$.

Pertaining to the decoding strategy, we could select the node with the maximum probability in a *greedy* manner or sample a node according to the probability in a *sampling* manner at each decoding step.

