# OpenReview forum: "Enhancing the Cross-Size Generalization for Solving Vehicle Routing Problems via Continual Learning"
_ICLR.cc/2024/Conference — Submitted to ICLR 2024_

### Official Review · Reviewer_SxzA · 2023-10-29

**Soundness:** 3 good
**Presentation:** 3 good
**Contribution:** 3 good
**Rating:** 6
**Confidence:** 2

**Summary:**

This work presents a continual learning-based framework to foster the cross-size generalization of deep models for VRPs. It leverages regularization schemes to retain the valuable insights derived from previously trained exemplar models to facilitate subsequent training. Experiments prove the effectiveness of the proposed method.

**Strengths:**

1. The proposed method is promising to use continual learning for vehicle routing problems.
2. The designed framework with inter-task regularization is reasonable.
3. This paper is well-presented with clear figures.

**Weaknesses:**

I'm not an expert in vehicle routing. And I list some concerns here for reference.

This work utilizes deep models and continual learning for vehicle routing problems. However, there is no detail on the adopted deep models in the paper. It makes me confused about how the framework in Figure 1 works.

**Questions:**

Please refer to the weakness section.

---

> ### Author Response · Authors · 2023-11-18
> **Response to Reviewer SxzA (Part 1/2)**
>
> Thank you very much for your time and effort in reviewing our work. We are glad to know you find our approach is promising, logical and well-presented. Please kindly note that we have made corresponding revisions to the paper, with the changes related to your questions marked in purple. Please refer to our updated version.
>
> We address your concerns as follows.
>
>
> &nbsp;
> ### Q1: Details of the adopted deep model.
>
> Thanks for raising this important concern. As shown in Section 3.2, we have briefly introduced the route construction process of prevailing deep models. For better readability and clarity, we have followed the suggestion and added the details of the adopted POMO model in the revised paper (refer to Section 4 / Paragraph 1 and Appendix E). Here we also present the details of POMO for your convenience.
>
>
> ### **E. Details of POMO model**
>
> POMO uses the encoder-decoder structure to sequentially construct solutions for VRPs. Specifically, the encoder projects problem-specific features into high-dimensional node embeddings for informative representation learning. Afterwards, the decoder sequentially constructs a solution $\tau= \begin{Bmatrix} \tau_1, \tau_2, ..., \tau_{T_c} \end{Bmatrix}$ with $T_c$ steps for a VRP instance of problem size $N$, conditioned on the node embeddings and partial tour at each step. Specifically, $T_c=N$ for TSP, and $T_c \geq N$ for CVRP as the depot node could be visited multiple times.
>
> **E.1 The Encoder**
>
> The encoder first embeds problem-specific features to higher-dimensional space, then passes them to stacked attention layers to extract useful information for better representation. The problem-specific features of node $v_i, i \in \begin{Bmatrix} 1,2,...,N \end{Bmatrix}$ contain 2-dimensional location coordinates (for both TSP and CVRP) and 1-dimensional demand vector (for CVRP only), which are linearly projected to initial node embedding $h_i^0$ of 128-dimension [1]. Then they are processed through $L=6$ attention layers with different parameters to derive the final node embedding $h_i^L$, where each attention layer is composed of a multi-head attention (MHA) sub-layer and a feed-forward (FF) sub-layer. Following the original design of the Transformer architecture [2], both the outputs of the MHA sub-layer and the FF sub-layer are followed by a skip-connection layer [3] and a batch normalization (BN) layer [4] as below,
>
> \begin{equation}
>     \tilde{h}_i = \text{BN}(h_i^l + \text{MHA}(h_i^l)),
> \end{equation}
> \begin{equation}
>     h_i^{l+1} = \text{BN}(\tilde{h}_i + \text{FF}(\tilde{h}_i)).
> \end{equation}
>
> **MHA sub-layer.** The MHA sub-layer employs a multi-head self-attention mechanism [2] with $M=8$ heads to compute the attention weights between each two nodes. Specifically, the *query/key/value* proposed in [2] are defined with $d_k = d / M$ dimension as below,
> $$q_i^{l,m}=W_Q^{l,m} h_i^l,\  k_i^{l,m}=W_K^{l,m} h_i^l, \ v_i^{l,m}=W_V^{l,m} h_i^l.$$
> Then the attention weights are computed by using the Softmax activation function to represent the correlation between each two nodes as follows,
> \begin{equation}
>     u_{ij}^{l,m} =\text{Softmax} \left(\frac{(q_i^{l,m})^{\top}(k_j^{l,m})}{\sqrt{d_k}}\right).
> \end{equation}
> Finally, the $l$-th MHA sub-layer first computes the new context vectors by performing an element-wise multiplication of the attention weights with *value*, and then aggregates the information from $M$ heads as follows,
> \begin{equation}
>     h_i^{l,m} = \sum_j u_{ij}^{l,m} v_j^{l,m}, \ m=1,2,...,M,
> \end{equation}
> \begin{equation}
>     \text{MHA}(h_i^l) = [h_i^{l,1}; h_i^{l,2}; ...; h_i^{l,M}] W^l_O,
> \end{equation}
> where $W_Q^{l,m}$, $W_K^{l,m}$, $W_V^{l,m} \in \mathbb{R}^{d \times d_k}$, $\ W_O^l \in \mathbb{R}^{md_k \times d}$ are learnable parameters, and $[;]$ denotes the concatenate operator.
>
> **FF sub-layer.** The FF sub-layer processes the node embeddings $\tilde{h}_i, i \in \begin{Bmatrix} 1,2,...,N \end{Bmatrix} $ through a hidden sub-layer with dimension 512 and a ReLU activation function, as follows,
> $$
> \text{FF}(\tilde{h}_i) = W_F^1 \ \text{ReLU}(W_F^0 \tilde{h}_i + b_F^0) + b_F^1,
> $$
> where $W_F^0, W_F^1, b_F^0, b_F^1$ are trainable parameters. At the end, the encoder outputs a set of node embeddings in the $L$-th layer $h_i^L, i \in \begin{Bmatrix}  1,2,...,N \end{Bmatrix} $. These embeddings will be preserved as a part of the input to the decoder for route construction.

---

> ### Author Response · Authors · 2023-11-18
> **Response to Reviewer SxzA (Part 2/2)**
>
> **E.2 The Decoder**
>
> Taking TSP as an example, the decoder first calculates the mean of node embeddings to derive the global graph embedding, i.e., $\bar{h} = \frac{1}{N} \sum_i^N h_i^L$, then defines a context vector as the combination of the graph embedding, the embeddings of the first node of the route and the last visited node. For step $t_c=1$, we use learned parameters $v^1$ and $v^2$ as input placeholders. Note that POMO constructs $N$ solutions by taking each of the $N$ nodes as the first node to visit (i.e., $h^i_{\tau_1} = h_i$). Particularly, it defines $N$ context vectors $h^c_i$, $\forall \ i \in \begin{Bmatrix} 1,2,...,N \end{Bmatrix}$ as follows,
> $$
> h^c_i=
> \begin{cases}
> (\bar{h}, h_i, h^i_{\tau_{t_c-1}}),  & \text{if}\ t_c>1 \\\\
> (\bar{h}, v^1, v^2), & \text{if}\ t_c=1
> \end{cases}
> $$
> where $h^i_{\tau_{t_c}}$ is the node embedding of the node visited at decoding step $t_c$ for the $i$th solution. The context vectors are then processed by a MHA layer as introduced above to generate $N$ glimpse vectors $h_i^{g}, i \in \begin{Bmatrix} 1,2,...,N \end{Bmatrix}$ in parallel,
> \begin{equation}
>     h_i^g = \text{MHA} (W^g_Q h^c_i,\ W^g_K h,\ W^g_V h), i \in \begin{Bmatrix} 1,2,...,N \end{Bmatrix},
> \end{equation}
> after which the decoder computes the compatibility between the enhanced glimpses and node embeddings. Then it further calculates the probabilities of selecting the next node to visit for $N$ solutions in parallel at decoding step $t_c$ as follows,
> \begin{equation}
>     c^{t_c}_i = G \cdot \text{tanh}\left(\frac{(h^g_i W_Q)^T (h W_K)}{\sqrt{d_k}}\right),
> \end{equation}
> \begin{equation}
>     p^{t_c}_i = \text{Softmax} (c^{t_c}_i), i \in \begin{Bmatrix} 1,2,...,N \end{Bmatrix},
> \end{equation}
> where $W^g_Q, W^g_K, W^g_V, W_Q, W_K$ are learnable parameter matrices, and $G$ is often set to 10 to control the entropy of $c^{t_c}_i$.
>
> Pertaining to the decoding strategy, we could select the node with the maximum probability in a *greedy* manner or sample a node according to the probability in a *sampling* manner at each decoding step.
>
> &nbsp;
>
> [1] W. Kool, H. van Hoof, and M. Welling. Attention, learn to solve routing problems! In International Conference on Learning Representations, 2018.
> [2] A. Vaswani, N. Shazeer, N. Parmar, J. Uszkoreit, L. Jones, A. N. Gomez, L. Kaiser, and I. Polosukhin. Attention is all you need. In Advances in Neural Information Processing Systems, pp. 5998–6008, 2017.
> [3] K. He, X. Zhang, S. Ren, and J. Sun. Deep residual learning for image recognition. In Proceedings of the IEEE Conference on Computer Vision and Pattern Recognition, pp. 770–778, 2016.
> [4] S. Ioffe and C. Szegedy. Batch normalization: Accelerating deep network training by reducing internal covariate shift. In International Conference on Machine Learning, pp. 448–456, 2015.

---

### Official Review · Reviewer_9M5K · 2023-10-31

**Soundness:** 3 good
**Presentation:** 3 good
**Contribution:** 3 good
**Rating:** 5
**Confidence:** 4

**Summary:**

This paper proposes a model-agnostic continual learning framework to improve the cross-size generalization capabilities of deep models for VRPs.
This paper designs the inter-task regularization scheme and intra-task regularization scheme to expedite the training on new sizes.
The proposed approach is evaluated on TSP and CVRP across a broad spectrum of sizes and shows good results.

**Strengths:**

1. This paper proposes the intra and inter task regularity, which is a main contribution of this paper, and shows the originality.
2. The proposed method sounds reasonable for continual learning.
3. The description of the whole method is relatively clear.
4. The experiments show good results and are relatively complete.

**Weaknesses:**

1. Although this paper proposes the intra and inter task regularity, the whole process is not novel enough.
It's very similar to the ema method.
2. There needs to be a clear conclusion on when to use intra-task regularization and when to use inter-task regularization, rather than just using "or".
3. It is better to $L_R$ instead of $L_{KR}$ for consistency in Line9 of Algorithm 1.

**Questions:**

1. Is this the first paper to deal with the cross-size generalization issue?
Is there any other way to solve the cross-size generalization problem?
The reviewers think the baselines are all for the single-sized problem, which may lose some important baselines.
2. What is the  $L_R$ if using the intra-task regularization in Eq.5? Please give a clear explanation.

---

> ### Author Response · Authors · 2023-11-18
> **Response to Reviewer 9M5K (Part 1/4)**
>
> Thank you very much for your time and effort in reviewing our work. We are glad to know you find our approach is logical and effective, has a clear description, and has sufficient and promising experiment results on cross-size generalization. Please kindly note that we have made corresponding revisions to the paper, with the changes related to your questions marked in red. Please refer to our updated version.
>
> Here we address your concerns as follows.
>
> &nbsp;
> ### W1. Novelty of our approach and its similarity to EMA.
>
> Thanks for pointing out this concern. Please note that the proposed regularization scheme is only one of our main contributions. We first show the differences between our approach and EMA, and then elaborate the contributions in detail.
>
> **Differences between our approach and EMA.**
>
> To our understanding, the EMA mentioned by the reviewer is most likely referring to the exponential moving average method, where the core of EMA is to assign a greater weight and significance on the most recent data points. We argue that our approach is different from EMA as follows.
> 1. **The whole framework.** Our primary goal is to develop a single model that delivers favorable performance over a broad spectrum of problem sizes, thus we train the model on all selected sizes with equal epochs, rather than only focusing on the most recent sizes.
> 2. **Collaboration of the experience replay and the regularization scheme.** Overemphasizing attention on recent (larger) sizes often lead to severe catastrophic forgetting, whereas excessively focusing on previously trained (smaller) sizes may lead to inadequate learning for current and more challenging tasks. To strike a balance, our approach incorporates both experience replay and a regularization scheme, which is different from EMA that focuses on the most recent tasks. The empirical results in Table 4 demonstrates the effectiveness of experience replay and regularization scheme.
> 3. **Regularization scheme.** Unlike EMA that simply assigns greater weights to the most recent tasks, our approach empowers the deep model to autonomously discern its focus by jointly optimizing the model using both task loss and regularization loss. Moreover, considering the NP-hard nature of VRPs [1], where the solving difficulty may exponentially increase as the problem scales up, we meticulously design the regularization scheme to transfer previously learned knowledge for facilitating the subsequent model training. The inter-task regularization updates the exemplar model after training on the entirety of each task, treating all sizes equally (in contrast to EMA that focuses primarily on the most recent tasks). While the intra-task regularization places additional emphasis on recent sizes, it specially offers a targeted solution to alleviate the NP-hard challenges inherent in VRPs.
>
> **Novelty of our approach.**
>
> We now elaborate our novelty and contributions in detail as below.
> 1. **Continual learning based framework.** Continual learning has emerged as a formidable approach for handling sequential tasks [2], and efficiently applying continual learning to domains such as image processing [3][4], natural language processing [5][6] and robotics [7][8], has garnered considerable attention and recognition. In comparison to these domains, solving VRPs poses heightened challenges due to its NP-hard nature. This complexity is further exacerbated in our specific problem setting that addresses the cross-size generalization issue, even including zero-shot generalization on unseen larger sizes. Considering the superiority of continual learning in sequential task learning, our approach is both **ingenious and novel**, as it strategically leverages the potential of continual learning to address our specific challenges. Notably, our work stands as the pioneering initiative to introduce continual learning into the VRP domain.
> 2. **Special designs of experience replay for VRPs.** A typical experience replay technique is to maintain a small memory buffer of training samples from past tasks and replay them during subsequent task training. To enhance memory efficiency, we dynamically generate instances of smaller sizes on the fly when training on a newly encountered larger size. Moreover, to prioritize the model training on the current task while retaining for well solving previous tasks, we employ mini-batch sampling. This is achieved by assigning a higher probability to selecting the current size and a lower probability to uniformly selecting one from previously encountered sizes. Despite its simplicity, this tailored experience replay strategy proves to be highly effective, as demonstrated in the revised Table 4 and Table 8.
> 3. **Regularization scheme.** Please kindly refer to "3. Regularization scheme" part in our response to "Difference between our approach and EMA".

---

> ### Author Response · Authors · 2023-11-18
> **Response to Reviewer 9M5K (Part 2/4)**
>
> ### W2. Selection of regularization scheme.
>
> Thanks for raising this valuable suggestion. We have followed it and added the selection strategy of intra-task and inter-task regularization schemes in the revised main paper (kindly refer to Section 4.2, Paragraph 2 and Paragraph 3). Specifically, we would like to clarify that:
>
> In our approach, the inter-task regularization scheme focuses on retaining knowledge derived from prior training on smaller sizes, whereas the intra-task one is tailored to consolidate recently learned knowledge on current larger size. Therefore,  the inter-task regularization scheme is favored for achieving generalization across various sizes, including (unseen) smaller ones. While the intra-task regularization scheme is the preferred option for generalizing to (unseen) larger sizes. According to our experiments, we observe that with comparable performance, the intra-task regularization is a bit more stable than inter-task regularization on TSP and CVRP. Hence we consistently utilize the intra-task regularization scheme to evaluate generalization performance on benchmark datasets TSPLIB and CVRPLIB in Table 3, as well as in all ablation studies, due to its superior performance in generalizing to larger sizes.
>
> Furthermore,  we explore the prospect of automating the regularization scheme selection during training based on task performance, and discuss it in the future work section of our revised paper (refer to Section 6). One potential solution involves dynamically adjusting the probability of selecting a scheme during training on each size $N_i$ ($i > 1$). This could be achieved by evaluating the model on the validation datasets of current task $N_i$ and previous tasks ($N_{pre} = {N_1, ..., N_{i-1}}$), and calculating gaps for these tasks relative to the LKH solver. If the model performs worse on the current size compared to previous ones, the intra-task regularization is automatically chosen; otherwise, the inter-task regularization is selected.
>
> &nbsp;
> ### W3. Presentation suggestion on revising $L_{KD}$ to $L_R$ in Line 9 of Algorithm 1.
>
> Thanks for pointing out the suggestion. We have followed the suggestion and revised accordingly. We will also meticulously review our paper to enhance its overall presentation.

---

> ### Author Response · Authors · 2023-11-18
> **Response to Reviewer 9M5K (Part 3/4)**
>
> ### Q1. Related works of cross-size generalization and baseline methods of our approach.
>
> Thanks for pointing out this concern. Please kindly note that enhancing cross-size generalization for VRPs holds significant importance and remains an area with limited exploration, which also serves as a primary motivation behind our work. Furthermore, our approach consistently outperforms existing methods that target cross-size generalization across all seen and unseen larger sizes for both TSP and CVRP.
>
> We now make following clarification: 1) summarize the related works of cross-size generalization for VRPs (responding to the first two sub-questions); 2) elaborate our baseline methods for enhancing cross-size generalization (responding to the last sub-question).
>
> **Related works of cross-size generalization.**
>
> As summarized in Section 2 (Related Works) in our paper, there are several works addressing cross-size generalization issue for VRPs. Notably, works like [9] and [10] are limited to TSP, lacking adaptability for broader VRP variants. Omni-POMO serves as a baseline in our paper, focusing on improving generalization performance cross sizes and distributions, achieving the state-of-the-art performance in cross-size generalization. Moreover, as introduced in Section 5.1 / Paragraph 1, AMDKD-POMO addresses cross-distribution generalization issue for VRPs. Given its inherent focus, we find it reasonable and logical to adapt AMDKD-POMO for cross-size generalization and include it as a baseline method. For a fair comparison, we retrain AMDKD-POMO according to our training setups, aligning its teacher models with our exemplar sizes. We also report the results of its open-sourced pre-trained models on the largest available size 100, i.e., AMDKD-POMO$^*$, showcasing the severe cross-size generalization issue of original AMDKD-POMO.
>
> **Cross-size generalization baselines.**
>
> As summarized above, both Omni-POMO and AMDKD-POMO focus on performing favorably **across multiple sizes instead of single ones**, and they are already integrated into our experiments as baseline methods (refer to Table 1 and Table 2). The results show that our approach consistently achieves lower average objective values than them across all sizes for both TSP and CVRP. Furthermore, we have added another baseline method designed for different-sized VRPs, i.e., **POMO-random** that trained on instances of random sizes within the same range [60, 150] as our approach in the revised Table 1 and Table 2. The results show that our approach significantly outperforms POMO-random across all seen and unseen sizes for both TSP and CVRP.
>
> In summary, we choose the competitive baselines from literature in experiments, including the state-of-the-art models like AMDKD-POMO and Omni-POMO that are specialized for generalization enhancement across multiple-sized problems, and the enhanced version of POMO trained on mixed sizes up to 150. Results show that our method can generally outperform all these baselines.

---

> ### Author Response · Authors · 2023-11-18
> **Response to Reviewer 9M5K (Part 4/4)**
>
> ### Q2: Definition of $L_R$.
>
> As shown in line 7 of Algorithm 1, $L_R$ is $L_{R_{intra}}$ defined in Eq. (4) if we use the intra-task regularization scheme, and is $L_{R_{inter}}$ defined in Eq. (3) if we use inter-task regularization scheme. For more clarity, we have further added the explanation of $L_R$ in the revised main paper (please kindly refer to page 6).
>
> &nbsp;
>
> [1] Lenstra J K, Kan A H G R. Complexity of vehicle routing and scheduling problems. Networks, 11(2): 221-227, 1981.
> [2] Hadsell R, Rao D, Rusu A A, et al. Embracing change: Continual learning in deep neural networks. Trends in Cognitive Sciences, 24(12): 1028-1040, 2020.
> [3] Zhou M, Xiao J, Chang Y, et al. Image de-raining via continual learning. In Proceedings of the IEEE/CVF Conference on Computer Vision and Pattern Recognition. pp: 4907-4916, 2021.
> [4] Zhai M, Chen L, Tung F, et al. Lifelong gan: Continual learning for conditional image generation. In Proceedings of the IEEE/CVF International Conference on Computer Vision. pp: 2759-2768, 2019.
> [5] Liu B, Mazumder S. Lifelong and continual learning dialogue systems: learning during conversation. In Proceedings of the AAAI Conference on Artificial Intelligence. 35(17): 15058-15063, 2021.
> [6] Srinivasan T, Chang T Y, Pinto Alva L, et al. Climb: A continual learning benchmark for vision-and-language tasks. In Advances in Neural Information Processing Systems. 35: 29440-29453, 2022.
> [7] Lesort T, Lomonaco V, Stoian A, et al. Continual learning for robotics: Definition, framework, learning strategies, opportunities and challenges. Information fusion. 58: 52-68, 2020.
> [8] Churamani N, Kalkan S, Gunes H. Continual learning for affective robotics: Why, what and how?. In IEEE International Conference on Robot and Human Interactive Communication. pp: 425-431, 2020.
> [9] Lisicki M, Afkanpour A, Taylor G W. Evaluating Curriculum Learning Strategies in Neural Combinatorial Optimization. In Advances in Neural Information Processing Systems, Workshop, 2020.
> [10] Zhang D, Xiao Z, Wang Y, et al. Neural TSP solver with progressive distillation. In Proceedings of the AAAI Conference on Artificial Intelligence, 37(10): 12147-12154, 2023.

---

### Official Review · Reviewer_h875 · 2023-10-31

**Soundness:** 3 good
**Presentation:** 2 fair
**Contribution:** 2 fair
**Rating:** 6
**Confidence:** 4

**Summary:**

The paper proposes a learning-based approach for vehicle routing problems that aims to learn models that generalize well across instances of different problem sizes. The method uses a continual learning scheme that sequential trains the model with instances of ascending sizes. Furthermore, the method uses the following new components to improve the generalization performance: 1) An inter-task regularization scheme and intra-task regularization scheme that aim to retain knowledge from earlier training phases. 2) An experience replay mechanism that revisits smaller instances from earlier training phases. The authors train models for the TSP and CVRP on instances with 60 to 150 nodes and evaluate them on instances with 60, 100, 150, 200, 300 and 500 nodes. The experiments show that their approach generalizes well to instances with 60-150 nodes. On larger instances, the approach offers only very minor improvements over a standard model that has been trained on instances with 150 nodes only.

**Strengths:**

- The approach seems to succeed in learning models that work well across instances of different sizes. On test instances with sizes seen during training, the trained models come close to the performance of POMO models trained specifically for one size only.
- The considered problem of improving the cross-size generalization performance of learning-based models is highly relevant.

**Weaknesses:**

- In my opinion the main weakness of this paper is that it does not provide enough ablation studies that evaluate the different components of the approach. The authors only provide some experiments for the TSP and only evaluate the experience replay and the intra-task regularization scheme in the main paper. In the Appendix some additional results for the TSP are reported but the CVRP is not considered at all. To fully convince me of the effectiveness of their proposed approach, the authors should report more detailed results for both problems in the main paper.
- I find Figure 1 and Figure 2 difficult to understand. At the time they are mentioned in the text multiple components shown in the figure have not yet been explained. Overall, I find the semantic of figure elements confusing. For example, what is the output of the “experience replay with previous tasks” element and why is there a connection between the current model and the “size selection” element? I think that Figure 1 especially should be simplified significantly.
- On larger test instances with sizes not seen during training the model leads to only small performance improvements over existing methods.
- The paper is missing a comparison to a naive POMO variant that is trained on instances of different sizes (e.g., in the range [60, 150]). The authors compare to AMDKD-POMO and Omni-POMO but both approaches seem to focus on the generalization to larger instances and do not perform well on the tasks reported in Table 1.

**Post-rebuttal:**
Thank you for your response and for providing additional results. My main concern (the lack of ablation studies) has been addressed and I raise my score accordingly. I still believe that the ablation study deserves more space in the main and that Figure 1 does not succeed in providing an easy to understand, high-level overview of the method.

**Questions:**

-

---

> ### Author Response · Authors · 2023-11-18
> **Response to Reviewer h875 (Part 1/4)**
>
> Thank you very much for your time and effort in reviewing our work. We are glad to know you recognize the high relevance and effectiveness of our approach in enhancing cross-size generalization performance. Please kindly note that we have made corresponding revisions to the paper, with the changes related to your questions marked in blue. Please refer to our updated version.
>
> We address your concerns as follows.
>
> &nbsp;
> ### W1. More ablation studies
>
> Thanks for raising this concern. In the revised paper, we have 1) enriched the ablation study on TSP in Table 4 and added ablation study on CVRP in Table 8 in Appendix C.1; 2) added ablation study on regularization schemes for CVRP in Figure 4 in Appendix C.2. Moreover, in Appendix D of the original submission, we have presented the validation regarding both seen and unseen sizes on TSP and CVRP. Please refer to Table 9 and its analysis in the revised appendix, which was Table 8 in the original submission.
>
> Please kindly understand that it is hard for us to put all experimental results into the main paper due to the page limit, we will try to give more clear direction and statement of those results in the revised main paper.
>
> **Ablation study on components of our approach for CVRP.**
>
> As suggested by the reviewer, in the revised Table 4 and Table 8, we consider all the components of our approach, i.e., the experience replay, inter-task regularization scheme and intra-task regularization scheme. Please note that only one regularization scheme (either inter-task regularization or intra-task regularization) is used at one time in our approach to keep a stable update of the exemplar model throughout the training. As shown in Table 4 and Table 8, both experience replay and the proposed regularization scheme (either inter-task one or intra-task one) contribute to the reduction of objective values and optimality gaps across all sizes for both TSP and CVRP. Further combining experience replay and regularization scheme together, both Ours-inter and Ours-intra (last two rows) achieve better performance, which demonstrates the effectiveness of each component in our approach.
>
> **Table 4: Ablation study on TSP.**
> | ER | Inter-task | Intra-task | N=60 |  | N=100 |  | N=150 |  |
> | :----: | :----:  | :----: | :----: | :----:  | :----: | :----: | :----: | :----: |
> | |  |  | obj. | Gap | obj. | Gap| obj. | Gap |
> | x | x | x | 6.1886 | 0.25% | 7.7898 |  0.32% | 9.3974 | 0.55% |
> | x | &#10003; | x | 6.1805  |  0.12% | 7.7831 | 0.24% | 9.3938 | 0.51% |
> | x | x | &#10003; | 6.1809 |  0.13% | 7.7829 |  0.24% | 9.3885 | 0.45% |
> | &#10003; | x | x | 6.1789| 0.10% | 7.7860 | 0.28%| 9.3932 | 0.50% |
> | &#10003; | &#10003; | x | 6.1758| 0.05% |7.7775 |0.17%|9.3883 |0.45%|
> | &#10003; | x | &#10003; | 6.1758| 0.05%| 7.7764 |0.15%|9.3820 | 0.38%|
>
> **Table 8: Ablation study on CVRP.**
> | ER | Inter-task | Intra-task | N=60 |  | N=100 |  | N=150 |  |
> | :----: | :----:  | :----: | :----: | :----:  | :----: | :----: | :----: | :----: |
> | |  |  | obj. | Gap | obj. | Gap| obj. | Gap |
> | x | x | x | 12.1174| 1.43%| 15.8528 |1.86% |19.5024 |2.35% |
> | x | &#10003; | x | 12.0829|1.14% |15.8109 | 1.59% | 19.4472 |2.06% |
> | x | x | &#10003; | 12.0855|1.16%|15.8085 |1.57% | 19.4289| 1.96%|
> | &#10003; | x | x | 12.0786 | 1.10%|15.8187|1.64%| 19.4389 |2.01% |
> | &#10003; | &#10003;| x |12.0660|1.00% |15.7848|1.42% | 19.4109 | 1.87%|
> | &#10003; | x | &#10003;|12.0663|1.00%| 15.7781 |1.37%|19.3938 |1.78%|
>
>
> &nbsp;
> **Ablation study on regularization schemes for CVRP.**
>
> Similar to Figure 3 that demonstrates the effectiveness of the proposed regularization schemes on TSP, we have added the results on CVRP as shown in Figure 4 in the revised paper (refer to Appendix C.2). As revealed, both inter-task and intra-task regularization schemes accelerate the learning efficiency and boost the overall results in comparison with the one without regularization scheme. Figure 3 and Figure 4 showcase the effectiveness of the proposed regularization schemes on both TSP and CVRP.

---

> ### Author Response · Authors · 2023-11-18
> **Response to Reviewer h875 (Part 2/4)**
>
> ### W2. Detailed description of Figure 1 and Figure 2
> Thanks for raising this important suggestion. For more clarity, we have modified Figure 1 and added more explanation to the caption for both Figure 1 and Figure 2 in the revised main paper. Specifically, we make the following clarification:
>
> **Regarding Figure 1**, for each mini-batch training of a task, we 1) first employ experience replay strategy to sample a size from formerly trained sizes and the current one using Eq. (2), so as to mitigate the catastrophic forgetting; 2) then generate mini-batch instances for the selected size; 3) finally use inter-task regularization to foster the current model to emulate an exemplary model for knowledge retention. These three steps are repeated for all mini-batch training during each task interval.
>
> **Pertaining to the reviewer's questions**, 1) the output of the “experience replay with previous tasks” element is a selected size, where $N_1$ is used as an example in Figure 1; 2) the connection between the current model and the “size selection” element aims to sample a new size using experience replay strategy for the next (new) mini-batch training.
>
> **Regarding Figure 2**, we illustrate the regularization scheme with two exemplar model updating strategies. (a) inter-task: exemplar model is updated after training on a whole task, which remains unchanged during training on a new task; (b) intra-task: exemplar model is updated multiple times during training on a task for concentrating more on newly encountered (larger) size.
>
> &nbsp;
> ### W3. Effectiveness of our approach on larger test instances
> Thanks for pointing out this concern. Please note that enabling a model to perform favorably across various sizes, and further achieving significant improvements for zero-shot generalization on unseen larger sizes is really difficult. This difficulty exists not only in our approach, but also in prevailing methods, such as [1-4].
>
> Regarding the comparison results on unseen larger sizes mentioned by the reviewer, we analyse them as follows:
>
> 1. **AMDKD-POMO:** In Table 2, both Ours-inter and Ours-intra achieve obvious superiority to AMDKD-POMO across three unseen larger sizes for both TSP and CVRP, e.g., with the optimality gap of 9.11% (Ours-intra) VS. 11.66% (AMDKD-POMO) for TSP500, and 7.70% (Ours-intra) VS. 9.78% (AMDKD-POMO) for CVRP500.
> 2. **Omni-POMO:** Omni-POMO is the most recent work addressing cross-size generalization issue for VRPs, where significant improvements over Omni-POMO might be difficult. Furthermore, Omni-POMO requires training on a broader range of sizes (i.e., [50, 200], including larger ones up to 200) than ours (i.e., [60, 150]), which inherently offers Omni-POMO the potential for superior performance on larger sizes. However, in Table 2, our approach still surpasses Omni-POMO on most of larger sizes, and Ours-intra achieves lower average objective values than Omni-POMO over the three larger sizes for both TSP and CVRP. This demonstrates the effectiveness of our approach on enhancing (zero-shot) cross-size generalization performance.
> 3. **POMO-150:** In the revised Table 2, we have added the results of *POMO-random*, i.e., training POMO on instances of random sizes within the range of [60, 150]. Notably, POMO-150, specifically trained on instances of the upper bound size 150, significantly outperforms POMO-random across all unseen larger sizes for both TSP and CVRP. This reveals that aligning the training size 150 more closely with unseen larger instances (i.e., 200, 300, and 500 in Table 2) enhances the potential for superior performance on larger sizes.
> Despite the superiority of POMO-150, Ours-intra trained on size range [60, 150] still outperforms POMO-150 across all larger sizes for both TSP and CVRP, further highlighting the effectiveness of our approach.
>
> In summary, the baselines in our experiments are fairly competitive, including the state-of-the-art models like AMDKD-POMO and Omni-POMO that are specialized for generalization enhancement, and the enhanced version of POMO trained on 150, as well as the ones trained on random sizes (i.e., POMO-random). Our approach consistently outperforms all baselines in cross-size generalization. Despite that the improvements on unseen instances seem not significant in comparison to those on seen instances, we reckon that these results are still encouraging and valuable, considering the inherent challenges of zero-shot generalization and the complexity of solving routing problems.

---

> ### Author Response · Authors · 2023-11-18
> **Response to Reviewer h875 (Part 3/4)**
>
> ### W4. Comparison baseline methods (Part one).
>
> **Compared to naive POMO trained on instances of different sizes:**
>
> Thanks for raising this valuable suggestion. Following it,  we have conducted comparison with naive POMO variant trained on instances of random sizes within the range of [60, 150], termed as *POMO-random*. The comparison results between POMO-random and our approach on both seen and unseen scales for TSP and CVRP are presented in Table 9 and Table 10. Both Ours-inter and Ours-intra significantly outperform POMO-random across all seen and unseen sizes for both TSP and CVRP, demonstrating the effectiveness of our approach.
>
> Please note that Omni-POMO also reports the POMO variants trained on various sizes in their paper, where POMO variants are inferior to Omni-POMO for both TSP and CVRP. Particularly, our approach showcases superiority to Omni-POMO in terms of cross-size generalization performance on both seen and unseen sizes as shown in Table 1 and Table 2. Hence we did not report POMO-random variant in original submission. However, for better readability and clarity, we have now added POMO-random variant into Table 1 and Table 2 in the revised main paper.
>
> **Table 9: Comparison results on TSP and CVRP (seen scales).**
>
> | Problem | Method |  |N=60  |  |  |N=100  |  | | N=150 | | Average of Total costs |
> | :----: | :----:  | :----: | :----: | :----:  | :----: | :----: | :----: | :----: |:----:|:----:|:----:|
> | TSP| | Obj. |Gap  | Time |Obj. |Gap  | Time |Obj. |Gap  | Time| |
> | | POMO-random | 6.1778 |0.08%| 36s |7.7782|0.18% | 2m|9.3937|0.51%| 33s|7.7832|
> | | Ours-inter | 6.1758| 0.05% |36s| 7.7775 | 0.17% | 2m |9.3883 | 0.45%  | 33s | 7.7805|
> | | Ours-intra | 6.1758 |0.05% | ～| 7.7764 | 0.15%  | ～ | 9.3820 |  0.38% | ～ | 7.7781|
> |CVRP| | | | | | | | | | | |
> | | POMO-random | 12.2758 | 2.75%  | 56s |15.7942 | 1.48%  | 3m | 19.6121 | 2.92%  | 33s |15.8940|
> | | Ours-inter | 12.0660| 1.00% | 56s | 15.7848 | 1.42%  | 3m | 19.4109 | 1.87%  | 33s |15.7539|
> | | Ours-intra | 12.0663 | 1.00%  | ~ | 15.7781|  1.37%  | ~|19.3938 | 1.78% |~ |15.7461|
>
> **Table 10: Comparison results on TSP and CVRP (unseen scales).**
> | Problem | Method |  | N=200 |  | |  N=300 |  ||  N=500  | | Average of Total costs |
> | :----: | :----:  | :----: | :----: | :----:  | :----: | :----: | :----: | :----: |:----:|:----:|:----:|
> | TSP| | Obj. |Gap  | Time |Obj. |Gap  | Time |Obj. |Gap  | Time| |
> | | POMO-random | 10.8397 | 1.61%  | 10s | 13.8212 | 6.70%  | 33s | 19.0881 | 15.53%  |2.5m |14.5830|
> | | Ours-inter | 10.7631 | 0.89% | 10s | 13.2942 | 2.63%  | 33s | 18.1047| 9.58%  | 2.5m |14.0540|
> | | Ours-intra | 10.7444 | 0.71%  | ~| 13.2263 | 2.11%  | ~| 18.0267|9.11%  |~|13.9992|
> |CVRP| | | | | | | | | | | |
> | | POMO-random | 23.2016 | 5.59% | 12s | 28.1393 | 8.89%  | 38s | 35.6822 | 14.99%  | 3m |29.0077|
> | | Ours-inter | 22.4847 | 2.33%  | 12s | 26.8134 | 3.76%  |38s |33.6337 | 8.39%  | 3m |27.6439|
> | | Ours-intra | 22.4436 |2.14% |~ | 26.6884| 3.28%  |~ | 33.4200 |7.70%  | ~ | 27.5173|

---

> ### Author Response · Authors · 2023-11-18
> **Response to Reviewer h875 (Part 4/4)**
>
> ### W4. Comparison baseline methods (Part two).
>
> **Compared to Omni-POMO and AMDKD-POMO:**
>
> Please note that both Omni-POMO and AMDKD focus on the performance **across multiple sizes** and/or distributions, instead of generalization to larger instances.
> **Regarding Omni-POMO**, as introduced in Section 2 / Paragraph 3, it is the most recent work for enhancing cross-size generalization performance for VRPs, instead of focusing on generalization to larger sizes. Although it seems to perform well on larger sizes as shown in Table 2 and their original paper, we find that such superiority actually sacrifices the performance on relatively small sizes as shown in Table 1. This compromise may be undesirable, since solving (extremely) large instances usually involves decomposing them into relatively small ones of different sizes, using some extra strategies like divide-and-conquer [4]. Furthermore, Omni-POMO is trained on a broader range of sizes (i.e., [50, 200], including larger ones up to 200) than ours (i.e., [60, 150]), which inherently offers Omni-POMO the potential for superior performance on larger sizes. However, in Table 2, our approach still achieves lower optimality gaps on most of larger sizes, and Ours-intra achieves lower averaged total cost than Omni-POMO across all sizes for both TSP and CVRP. These results underscore the superiority of our approach on enhancing cross-size generalization performance.
> **Regarding AMDKD-POMO**, as introduced in Section 5.1 / Paragraph 1, it aims at enhancing cross-distribution generalization for VRPs. Given its inherent focus, we find it reasonable and logical to adapt AMDKD-POMO to address cross-size generalization issue and include it as a baseline method. For fair comparison, we retrain AMDKD-POMO following our training setups by tailoring its teacher models to align with our exemplar sizes. We also show the results of its open-sourced pretrained models on the largest available size 100, i.e., AMDKD-POMO$^*$, to demonstrate the severe cross-size generalization issue of original AMDKD-POMO. Particularly, both Ours-inter and Ours-intra surpass AMDKD-POMO$^*$ and AMDKD-POMO across both seen and unseen larger sizes for both TSP and CVRP as shown in Table 1 and Table 2.
>
> &nbsp;
>
> [1] Zhou J, Wu Y, Song W, et al. Towards Omni-generalizable Neural Methods for Vehicle Routing Problems. In International Conference on Machine Learning, 2023.
> [2] Lisicki M, Afkanpour A, Taylor G W. Evaluating Curriculum Learning Strategies in Neural Combinatorial Optimization. In Advances in Neural Information Processing Systems, Workshop, 2020.
> [3] Qiu R, Sun Z, Yang Y. Dimes: A differentiable meta solver for combinatorial optimization problems. In Advances in Neural Information Processing Systems, 35: 25531-25546, 2022.
> [4] Hou Q, Yang J, Su Y, et al. Generalize Learned Heuristics to Solve Large-scale Vehicle Routing Problems in Real-time. In the Eleventh International Conference on Learning Representations. 2022.

---

### Author Response · Authors · 2023-11-18
**General Response**

To begin with, we thank all reviewers for their valuable comments on our paper. Please kindly note that we have made corresponding revisions to the main paper and appendix based on the comments, as presented in our updated version.. We provide author responses one by one.

---

### Author Response · Authors · 2023-11-21
**General Response**

Dear reviewers:
    Thanks again for your efforts in reviewing our work. Since there are only 2 days left to the discussion deadline, we could want to know whether all of your concerns have been properly addressed. We are more than happy to further address any remaining concerns you might have.

---

### Meta-Review · Area_Chair_SFcW · 2023-12-06

**Metareview:**

The paper studies a deep-learning-based heuristics for vehicle routing problems. It builds on ideas in prior work, that often train for fixed problem sizes. The proposed approach uses a continuous learning approach to regularize solutions between different problem sizes (intra-task) and within a specific problem size (inter-task).
The paper was reviewed by three experts in the field, all three have borderline recommendations (two borderline accept, one borderline reject).

+ The approach shows promising results, especially across problem sizes and on novel sizes
+ The solution is reasonable
- As pointed out by Reviewer 9M5K (Q1) the baselines are slightly artificial. For example, besides a computational overhead, what is wrong with training and solving a TSP for all three sizes and picking the best result? Instead the authors restrict themselves to models trained with fixed or randomizes sizes. The paper would clearly be stronger if the authors did show that slightly more challenging baselines do not generalize as well as the proposed method. The new results presented in the discussion hint at this, but are not sufficient to make a strong enough case.
- Some of the presentation, especially figure 1, could use more work, even after the revision.

Overall, the paper is on a good track, but not quite at the level of an ICLR paper. The discussion hinted at some promising new results, but working these into the presentation and story would require some significant changes in the presentation and analysis. This in turn would likely require reviewers to take another look (re-submission).

**Justification For Why Not Higher Score:**

The paper does show promise, but would require some significant changes in the way the approach is motivated and comparisons/ablations are worked in to be accepted.

**Justification For Why Not Lower Score:**

N/A

---

### Decision · Program_Chairs · 2024-01-16

Reject